# Development and calibration of an automatic spectral albedometer to estimate near-surface snow SSA time-series

G. Picard[1,2], Q. Libois[1 a], L. Arnaud[1], G. Verin[1], and M. Dumont[3]

[1]UJF – Grenoble 1 / CNRS, Laboratoire de Glaciologie et Géophysique de l'Environnement (LGGE) UMR 5183, Grenoble, F-38041, France

[2]ACE CRC, University of Tasmania, Private Bag 80, Hobart, TAS 7001, Australia

[a]now at: ESCER Centre, Department of Earth and Atmospheric Sciences, Université du Québec à Montréal (UQAM), Montréal, Canada

[3]Météo-France – CNRS, CNRM – GAME UMR 3589, Centre d'Études de la Neige, Grenoble, France

*Correspondence to:* Ghislain Picard (ghislain.picard@ujf-grenoble.fr)

**Abstract.** Spectral albedo of the snow surface in the visible/near-infrared range has been measured for 3 years by an automatic spectral radiometer installed at Dome C (75°S, 123°E) in Antarctica in order to retrieve the specific surface area (SSA) of superficial snow. This study focuses on the uncertainties of the SSA retrieval due to instrumental and data processing limitations. We find that when the solar zenith angle is high, the main source of uncertainties is the imperfect angular response of the light collectors. This imperfection introduces a small spurious wavelength-dependent trend in the albedo spectra which greatly affects the SSA retrieval. By modeling this effect, we show that for typical snow and illumination conditions encountered at Dome C, retrieving SSA with an accuracy better than 15% (our target), requires the difference of response between 400 and 1100 nm not to exceed 2%. Such a small different can be achieved only by i) a careful design of the collectors, ii) an *ad hoc* correction of the spectra using the actual measured angular response of the collectors, and iii) for solar zenith angles less than 75°. The 3-year long time-series of retrieved SSA features a three-fold decrease every summer which is significantly larger than the estimated uncertainties. This highlights the high dynamics of near-surface SSA at Dome C.

## 1 Introduction

The summer surface energy budget on the ice sheets is largely controlled by the absorption of solar energy by snow (Van As et al., 2005; Ettema et al., 2010). The latter is determined by the downward irradiance reaching the surface and by snow albedo. With solar irradiance of several hundreds of $W\,m^{-2}$ during daylight, a persistent change of 1% of the albedo represents an energy comparable with the globally averaged radiative forcing caused by $CO_2$ concentration increase since pre-industrial time (1.82 $W\,m^{-2}$, Myhre et al., 2009). While planetary albedo change is not expected to exceed a 0.1% (Myhre et al., 2009), local changes can be much larger owing to the dependence of snow albedo on multiple factors including snow grain size and shape, surface roughness, snow depth, and the amount of light-absorbing impurities such as black carbon, dust and biological pigments (e.g. Warren and Wiscombe, 1980; Warren et al., 1998; Aoki et al., 2000; Dumont et al., 2010; Zhuravleva and Kokhanovsky, 2011; Stibal et al., 2012; Goelles et al., 2015). These factors vary in space and time depending on the

atmospheric conditions and are controlled by numerous processes giving rise to complex snow-albedo feedback loops between the snow cover and the atmosphere (Curry et al., 1995; Qu and Hall, 2007; Picard et al., 2012). In Antarctica, snow grain size is the main factor controlling the albedo along with the illumination conditions (solar zenith angle and cloud cover) and surface roughness while the impurity content is usually low (Warren et al., 2006). Investigating the feedback loops thus requires
long-term and accurate observations of albedo and snow properties.

Nevertheless, albedo measurements in Antarctica are scarce and subject to artifacts. Only a few Baseline Surface Radiation Network (BSRN) stations and some automatic weather stations deployed by the Institute of Marine and Atmospheric research (Utrecht University) provide time series of broadband albedo measured using upward and downward looking pyranometers. In addition, these sensors are subject to many artifacts such as imperfect response at high solar zenith angle, leveling and frost,
which are difficult to correct (van den Broeke et al., 2004; Bogren et al., 2016). Retrieving broadband albedo from remote sensing is not a straightforward alternative. It requires atmospheric correction, spectra extrapolation from a limited number of spectral bands, and angular extrapolation of the bi-conical measurements to get hemispherical surface reflectance (Stroeve et al., 2006). The latter is challenging because the bi-directional reflectance of snow is much more sensitive to grain shape (Dumont et al., 2010) and surface roughness (Kuchiki et al., 2011) than hemispherical reflectance (i.e. albedo), thus introducing extra
unknown variables that need to be estimated somehow. Moreover, satellite data can not be used under cloudy conditions or when the sun zenith angle is too high (Wang and Zender, 2010; Schaaf et al., 2011). Despite many satellite overpasses per day in polar regions, the time-series are discontinuous and snowfalls, which are periods of great albedo change, are systematically missed.

Measuring snow grain size is difficult as well. Estimating the maximal extent of the dominant crystals using hand-lens
or binocular is the most widely used technique to measure grain size (Fierz et al., 2009; Aoki et al., 2000; Pirazzini et al., 2015). However, not only is it known to be imprecise and operator-dependent, but also it can not be automated for unmanned monitoring of grain size evolution. Recent advances in field measurement of the specific surface area (the surface area of the air-ice interface per unit of mass of snow) – which is equivalent to the optical radius commonly used in remote sensing (Grenfell and Warren, 1999; Nolin and Dozier, 2000; Jin et al., 2008) – offer an attractive alternative. Examples of instruments
and methods include contact spectroscopy (Painter et al., 2007), near-infrared photography (Matzl and Schneebeli, 2006), DUFISSS and IceCUBE (Gallet et al., 2009) and POSSSUM and ASSSAP (Arnaud et al., 2011). These techniques being based on optical measurements of the snow reflectance in the near or short wave infrared, it is arguable whether they provide data more related to albedo or a proper geometrical metric of the snow micro-structure. Several theoretical studies have indeed shown the significant influence of the snow crystal shape on the albedo (Macke and Mishchenko, 1996; Neshyba et al., 2003;
Kokhanovsky and Zege, 2004; Picard et al., 2009) and on the e-folding depth (Libois et al., 2013) but no clear experimental evidence of this effect has been given yet for snow on the ground (Gallet et al., 2009; Libois et al., 2014b). In fact, these optical techniques have been shown to be accurate within 15% when compared with independent measurements using methane adsorption or micro-computed tomography techniques (Matzl and Schneebeli, 2006; Domine et al., 2006; Gallet et al., 2009). This accuracy is sufficient to achieve a 1% accuracy of broadband albedo (Gardner and Sharp, 2010) which is considered to
be adequate for climate study (Bogren et al., 2016). Intensive campaigns of SSA measurements have provided new insight

of the snow metamorphism in the Alps and in Antarctica over seasons and have allowed refined evaluation of detailed snow model predictions (Picard et al., 2012; Morin et al., 2013; Roy et al., 2013; Libois et al., 2015). However, these techniques still require an operator which is often a limitation, particularly in the Antarctic. Optical satellite remote sensing has been used to retrieve optical radius (e.g. Nolin and Dozier, 2000; Painter et al., 2009; Negi et al., 2011; Mary et al., 2013) in particular in Antarctica (Scambos et al., 2007; Jin et al., 2008). Mary et al. (2013) and Tanikawa et al. (2015) give an overview of the numerous existing algorithms. Obtaining grain size on a large spatial extent and for many years is of great interest, but attaining an accuracy similar to that of field techniques requires to overcome several additional defects that are specific to space-borne sensors and are in fact very similar to those faced for albedo retrieval. Microwave remote sensing is not subject to these issues but as far as albedo study is concerned, only the highest available frequencies could be able to provide superficial grain size (Picard et al., 2012). However, the technique proposed by Picard et al. (2012) is specific in many ways to the conditions of the Antarctic Plateau and requires additional assumptions about the density and temperature of superficial snow compared to the optic domain.Wuttke et al. (2006)

Ground-based observation of spectral albedo has several advantages compared to broadband measurement: it can provide a finer understanding of the albedo variation and can be used to estimate SSA using similar principles to those used in remote sensing and by the optical techniques mentioned above. It is also easier to assess the data quality or develop corrections by exploiting the spectral signature. Many spectral albedo measurements have been conducted in Antarctica (e.g. Grenfell et al., 1994; Hudson et al., 2006; Hudson and Warren, 2007; Marks et al., 2015; Pirazzini et al., 2015) but they were limited to short term campaigns and the temporal evolution was not the priority. Long-term monitoring of spectral albedo requires fully automatic instruments with a good robustness to cope with the Antarctic conditions (Wuttke et al., 2006; Salzano et al., 2016). With the advent of cheaper and compact spectrometers free of mechanical parts (Kantzas et al., 2009; Nicolaus et al., 2010), operating such instruments in Antarctic conditions is becoming easier.

The objective of the present study is to describe the automatic spectral radiometer that we developed and installed at Dome C (75°S, 123°E) and the data processing developed to estimate time-series of spectral albedo and SSA from the raw irradiance measurements. We particularly focus on the uncertainties of the estimated albedo and SSA. For this, we analyze the improvement of the quality through the successive processing steps. We discuss the results in the light of a SSA target uncertainty of 15%, as claimed to be reached by manual devices – near-infrared photography (Matzl and Schneebeli, 2006), DUFISSS (Gallet et al., 2009), POSSSUM (Arnaud et al., 2011)) – and considering it is difficult to perform better than those man-operated tools. Theoretical modeling of the imperfections of the instrument are used to help in this analysis and also to provide quantitative recommendations for the design of future spectral radiometers. Simulations are also used to explore the representativeness of the retrieved SSA. The geophysical interpretation of the 3-year long SSA time-series obtained with our instrument and its comparison to snow model simulations are addressed in Libois et al. (2015).

Section 2 describes the retrieval of SSA from spectral albedo and uncertainties induced by instrumental artifacts. Section 3 presents the instrument deployed at Dome C and the data processing to obtain calibrated albedo. The 3-year long time-series of retrieved SSA is presented and analyzed in Section 4. Discussion and conclusion are provided in the last section.

## 2 Theory

In this section, we present the algorithm to retrieve SSA from observed spectral albedo and evaluate the uncertainties resulting from working hypothesis and instrumental artifacts.

### 2.1 Retrieval of specific surface area from spectral albedo

Snow SSA at the surface is obtained by fitting a theoretical albedo model with SSA as the main unknown to observed albedo spectra over some range of wavelengths. To keep the number of unknowns moderate in order to prevent unstable optimization, the following assumptions are made: 1) The snowpack is horizontally and vertically homogeneous which means only one SSA value is retrieved. 2) The surface is flat. Roughness which tends to weaken the solar zenith angular response of the snow is neglected (Warren et al., 1998). 3) The surface and the sensor are perfectly horizontal. 4) Snow phase function and single scattering albedo are implicitly described by the asymmetry factor, absorption enhancement parameter, and SSA (Kokhanovsky and Zege, 2004). The values of asymmetry factor and absorption enhancement parameter are assumed constant and are taken from Libois et al. (2014b). 5) Snow is clean or contains sufficiently small quantities of impurity not to impact the albedo in the considered wavelength range. This is usually the case in Antarctica (Grenfell et al., 1994).

In these conditions, the analytical asymptotic radiative transfer (ART, Kokhanovsky and Zege, 2004; Negi et al., 2011) is valid and gives the directional and diffuse hemispheric reflectances $\alpha^{\text{diff}}$ and $\alpha^{\text{dir}}$ as a function of SSA:

$$\alpha^{\text{diff}}(\lambda) = \exp\left(-4\sqrt{\frac{2B\gamma(\lambda)}{3\rho_{\text{ice}}\text{SSA}(1-g)}}\right) \tag{1}$$

$$\alpha^{\text{dir}}(\lambda,\theta) = \exp\left(-\frac{12}{7}(1+2\cos\theta)\sqrt{\frac{2B\gamma(\lambda)}{3\rho_{\text{ice}}\text{SSA}(1-g)}}\right), \tag{2}$$

where $\theta$ is the solar zenith angle, $\rho_{\text{ice}} = 917\,\text{kg}\,\text{m}^{-3}$ is the ice density at $0\,°\text{C}$, and $\gamma(\lambda)$ is the absorption coefficient of ice, taken from Warren and Brandt (2008). $B = 1.6$ and $g = 0.85$ are respectively the absorption enhancement parameter and the asymmetry factor values suggested by Libois et al. (2014b).

Considering that observed albedo is composed of a direct and diffuse component, the model reads:

$$\alpha^{\text{1-param}}(\lambda,\theta) = \left[r^{\text{diff}}(\lambda,\theta)\alpha^{\text{diff}}(\lambda) + \left(1-r^{\text{diff}}(\lambda,\theta)\right)\alpha^{\text{dir}}(\lambda)\right], \tag{3}$$

where $r^{\text{diff}}(\lambda,\theta)$ is the ratio of diffuse over direct irradiance and is supposed to be known. This simple model is called hereinafter 1-parameter model because SSA is the only unknown. However, it is not the most suitable because albedo measurements are subject to various artifacts such as: variations of the illumination or poor cross-calibration between the downwelling and upwelling measurements, slopes of the surface, etc, sometimes even resulting in values larger than 1 (Pirazzini et al., 2015). To cope with artifacts that do not depend on wavelength, a free wavelength-independent scaling parameter $A$ is introduced:

$$\alpha^{\text{2-param}}(\lambda,\theta) = A\left[r^{\text{diff}}(\lambda,\theta)\alpha^{\text{diff}}(\lambda) + \left(1-r^{\text{diff}}(\lambda,\theta)\right)\alpha^{\text{dir}}(\lambda)\right], \tag{4}$$

This 2-parameter model is more flexible but to avoid over-fitting, we discard spectra that would result in $A$ out of the range 0.9–1.1.

The fit is performed with a non-linear least square method (provided by the Python scipy.optimize.leastsq function) to minimize the squared difference between the model and observed spectrum.

The choice of wavelength range to perform the fit depends on the available observation. In our case, spectrometers with Silicium sensor are used so only data at wavelengths shorter than 1050 nm present a sufficient signal/noise ratio. In addition, we restrict the fit to wavelengths longer than 700 nm because the sensitivity of the albedo to SSA is stronger in the near-infrared than in the visible, and impurities are known to affect visible wavelengths (France et al., 2011).

## 2.2   Vertical representativeness of retrieved SSA values

Assuming the snowpack is vertically homogeneous and retrieving a single bulk SSA stabilizes the fit. However, this implies to interpret the retrieved value with caution. There are indeed some evidences of significant vertical gradient in the topmost centimeter in Antarctica. Grenfell et al. (1994) found at South Pole that adding a thin layer of very small snow grains (SSA larger than 100 $\mathrm{m^2\,kg^{-1}}$) on the surface in their radiative transfer calculations was required to match their albedo measurements. They attributed this to the deposition of snow particles having been shrunk by sublimation during blowing snow events.

Carmagnola et al. (2013) at Summit in Greenland use spectral data in the visible, near and short wave infrared and found that tuning the surface layer SSA helped to improve the agreement with the observations but did not explore values as high as Grenfell et al. (1994) did. In contrast, Gallet et al. (2011) measured SSA vertical profiles at Dome C and obtained radiative transfer calculations in agreement with Hudson et al. (2006) observations without adding such a layer. Using snowpack numerical modeling, Libois et al. (2015) obtained SSA profiles at Dome C with such a significant gradient near the surface. This

is particularly remarkable as their simulations do not consider some processes susceptible to bring small grains on the surface such as shrinking of air-borne particles by sublimation or clear-sky precipitation (a.k.a diamond dust).

Because of this gradient, it is important to know over which thickness the retrieved SSA is representative. Unfortunately, there is no simple answer because the penetration depth of radiation in snow depends not only on the wavelength but also on the vertical profile of snow properties, including SSA itself and density, which are both unknown. Furthermore, the estimated

SSA value is not a simple average over a given thickness but is a weighted average with a kernel function decreasing with depth. This function has an exponential form for homogeneous snowpack but is more complex otherwise.

Only multi-layer radiative transfer calculations can give a precise estimation of the sensitivity-depth relationship when SSA and density profiles are known. To evaluate this relationship in the simple case of an homogeneous snowpack, we proceed as follows: the two-stream radiative transfer model TARTES (Libois et al., 2013) is run considering a semi-infinite medium with

fixed SSA and a density of 270 $\mathrm{kg\,m^{-3}}$ (typical mean surface conditions at Dome C, Libois et al. (2014a)). A layer with variable thickness $h$ is added on top of it, with the same properties, excepted that a small perturbation $\delta$SSA is added to the SSA. The relative contribution of the uppermost layer to the albedo is then defined as the quantity $(\alpha(h) - \alpha(0))/(\alpha(\infty) - \alpha(0))$ where $\alpha(h)$ is the albedo calculated by TARTES for the thickness $h$ and averaged over the wavelength range used by Autosolexs (700-1050 nm). The actual value of $\delta$SSA does not change the final result owing to the normalization, as long as it is small

(e.g. $+1\,\mathrm{m^2\,kg^{-1}}$). The contribution is shown in Figure 1 (plain line) as a function of $h$ for different SSA values. Results show that the uppermost 10 mm snow layer contributes to nearly 60% of the albedo for SSA of $20\,\mathrm{m^2\,kg^{-1}}$ and 85% for higher SSA of $100\,\mathrm{m^2\,kg^{-1}}$. Conversely, the layer contributing to 80% of the signal is 18, 12 and 8mm thick for SSA of 20, 50 and 100 respectively. We can conclude that $1\,\mathrm{cm}$ is a reasonable approximation for Dome C conditions which gives a more quantitative meaning to the term "near-surface" used throughout this article.

For comparison, we also run similar simulations at $1310\,\mathrm{nm}$, a wavelength commonly used by device to measure SSA (Gallet et al., 2009; Arnaud et al., 2011) (dashed line in Figure 1). In this case, most of the signal comes from the uppermost 5 mm in any common conditions. Note that the penetration depth is inversely proportional to density so that the thickness values presented here should be multiplied by about two in case of fresh snow or surface hoar.

## 2.3 SSA uncertainty due to instrumental artifacts

Even when the assumptions listed in Section 2.1 are met, the SSA retrieval can be impacted by observation imperfections, such as dark current of the radiometer, leveling and angular response of the light collector, calibration, etc. Here, such impacts are investigated using a simple theoretical framework. Most of these effects can be represented in a first approximation as either a bias in the measured irradiances (hereinafter called offset) or a small spurious wavelength-dependent trend in the spectra (hereinafter called chromatic aberration). We investigate theoretically these two cases successively in the next subsections and apply the result to real observations in Section 3. In both cases, to quantify the impact on the estimation of the SSA, we perform simple and idealized numerical experiments as follows: we first compute the perfect albedo spectrum with the model described in Equation 1 for a given SSA (called true SSA) and perturb it to mimic the considered artifact. Then, we retrieve the SSA using the algorithm described in Section 2.1 and deduce the error as the difference between the estimated and true SSA.

### 2.3.1 Offset

We consider the true reflected and incident radiance spectra $S^{\mathrm{ref}}(\lambda)$ and $S^{\mathrm{inc}}(\lambda)$ are affected by a constant bias $d$. The measured albedo writes:

$$\alpha(\lambda) = \frac{S^{\mathrm{ref}}(\lambda) + d}{S^{\mathrm{inc}}(\lambda) + d}. \tag{5}$$

Dividing by the incident radiance makes explicit the true albedo $\alpha^{\mathrm{true}}$:

$$\alpha(\lambda) = \frac{\alpha^{\mathrm{true}} + \frac{d}{S^{\mathrm{inc}}(\lambda)}}{1 + \frac{d}{S^{\mathrm{inc}}(\lambda)}}. \tag{6}$$

For the incident spectrum $S^{\mathrm{inc}}(\lambda)$, we choose a Gaussian shape looking like the observations (Section 3.3):

$$S^{\mathrm{inc}}(\lambda) = S_{\mathrm{mode}} \exp\left[-\left(\frac{\lambda - 680\mathrm{nm}}{270\mathrm{nm}}\right)^2\right]. \tag{7}$$

The amplitude of the spectrum $S_{\mathrm{mode}}$ is usually of the order of the resolution of the Digital Analog Converter of the radiometer ($2^{16}$ in our case) if the integration time is optimal. We define $d' = d/S_{\mathrm{mode}}$, the relative contribution of the offset with respect to

the spectrum amplitude. We then estimate the SSA from the perturbed $\alpha(\lambda)$ spectrum and calculate the difference with respect to the true SSA.

Figure 2 shows the SSA error as a function of the relative offset $d'$ ranging from 0 (perfect instrument) to 2% for several SSA values. The result is nearly independent of the true SSA. To meet the 15% accuracy criteria, the offset needs to remain under $d'$=1%. For a spectrometer, this offset can generally be estimated using dark pixels (pixels receiving any light so that $S(\lambda)=0$). Since their signal-to-noise ratio is usually much better than 1:100, the offset correction should not in most case be an obstacle to achieve the targeted accuracy.

### 2.3.2 Chromatic aberration

To investigate the impact of chromatic aberration, we consider the true albedo $\alpha^{\text{true}}$ is multiplied by a linear function of the wavelength:

$$\alpha(\lambda, \theta) = \left(1 - b\frac{\lambda - \lambda_0}{\lambda_1 - \lambda_0}\right)\alpha^{\text{true}}(\lambda, \theta) \tag{8}$$

where $b$ controls the slope of the wavelength dependence. The limits $\lambda_0$=400 nm and $\lambda_1$=1100 nm are chosen so that the instrument is assumed perfect at 400 nm in the blue and $b$ represents the relative response difference between the upward and downward looking channels at 1100 nm. Figure 3 illustrates the results with true SSA of 50 $\text{m}^2\,\text{kg}^{-1}$ and $b = 0.05$ (actual values of $b$ are given in this section and in Section 3). We remind that the SSA estimation only uses data between 700–1050 nm. The retrieved SSA is 38.2 $\text{m}^2\,\text{kg}^{-1}$ with the 2-parameter model and 27.6 $\text{m}^2\,\text{kg}^{-1}$ with the 1-parameter one. This corresponds to relative errors of 24% and 45% respectively. The 2-parameter model is clearly better which can be explained by the fact that it is insensitive to the scaling of the albedo and depends mostly on the spectal variations of albedo within the range used for the optimization. However, this error is larger than the 15% target accuracy in SSA.

An interesting feature of the 2-parameter model is that the albedo predicted with the optimal SSA and $A$ values (orange curve) exhibits an offset in the visible not only with respect to the true albedo (black curve) but also to the observed one (green curve). For instance, in Figure 3, the difference is 0.024 on average over the range 400–550 nm. This difference (hereinafter referred to as residuals) is a consequence of the wavelength-dependent perturbation and can be exploited to assess the amplitude of this perturbation in real data because it does not require knowledge of the true albedo. It means that acquisitions subject to residual chromatic aberration feature a large difference in the visible range with respect to the spectrum predicted using the 2-parameter model. To explore how this difference and the relative error on retrieved SSA depend on actual SSA, Figure 4a shows the relative error and the mean difference over the range 400–550 nm as a function of the chromaticity parameter $b$ for several SSA. The error varies almost linearly with $b$ and the slope increases with SSA. Hence, to attain the target accuracy of 15%, the chromatic parameter should not exceed about 0.04 for SSA=20 $\text{m}^2\,\text{kg}^{-1}$ and remain under 0.02 for SSA=100 $\text{m}^2\,\text{kg}^{-1}$. Figure 4b shows the nearly univocal relationship between $b$ and the mean residual. This suggests that mean residual values under 0.01 indicate a chromaticity $b$ better than 0.02 and thus an SSA accuracy better than 15%, thereby providing a fast and simple spectra quality check criteria.

We implemented this criteria in our algorithm as follows: 1) predict the albedo spectrum with the 2-parameter model using the optimal SSA and $A$ parameters obtained by fitting the model to the observed spectrum in the range 700–1050 nm. 2) calculate the mean difference in the range 400–550 nm and 3) reject the spectrum if this difference is larger than 0.01. This filter works at Dome C where snow is clean, but would not work in the presence of light absorbing impurities because the latter

cause a decrease of the albedo in the blue-green domain (Warren, 1982) which would result in systematic rejection by the filter.

Several artifacts results in chromatric aberration. Leveling of the collector is one of them. Tilt angle $\delta$ of the collector with respect to the horizontal indeed tends to affect albedo by a scaling $\eta(\lambda, \theta)$ that can be written according Bogren et al. (2016) and Grenfell et al. (1994):

$$\eta(\lambda, \theta) \approx \left(1 - r^{\mathrm{diff}}(\lambda, \theta)\right) \left(\frac{\cos(\theta - \delta)}{\cos(\theta)} - 1\right) \tag{9}$$

where the diffuse term error has been neglected as suggested by Bogren et al. (2016) findings and tilt in the direction of the sun (worse case) is considered. Assuming the errors are small enough to be treated as perturbations, $b$ can be deduced as follows:

$$b \approx \eta(\lambda_1, \theta) - \eta(\lambda_0, \theta) \tag{10}$$

$$\approx \left(r^{\mathrm{diff}}(\lambda_1, \theta) - r^{\mathrm{diff}}(\lambda_0, \theta)\right) \left(\frac{\cos(\theta - \delta)}{\cos(\theta)} - 1\right) \tag{11}$$

For a tilt angle of 1° and under typical conditions for Dome C, we obtain $b = 0.01$ for SZA=53° (as the noon acquisition

studied in Section 3.4) and $b = 0.05$ for SZA=77° (as the evening acquisition) which is weak in the former case but is too high in the latter one to reach the 15% target accuracy. These results highlight the crucial role of the leveling of the instrument. Other defects resulting in chromatic aberration are addressed in Section 3.4 on practical examples.

## 3    Measurements of spectral albedo

A specific instrument has been developed to measure the spectral albedo of the surface at Dome C. The following subsec-

tion describes the instrument, details on the construction and characterization of the light collectors (or fore optics) and data processing steps.

### 3.1    Autosolexs instrument

We designed and assembled an instrument composed of several commercial and home-made components to automatically measure spectral albedo. The main specifications driving the design were the robustness to work in Dome C harsh environment

for several years and the ability to acquire not only albedo (incident and reflected radiation) but also the radiation within the snow at several levels (not used in this study), thus requiring a spectrometer with many inputs. The instrument is named Autosolexs (Automatic SOlar Extinction in Snow) because of the latter application.

Figure 5 shows pictures of the visible above-ground part of the system which comprises two heads for albedo measurements at about 2 m above the surface. Due to snow accumulation, this height has decreased at an averaged rate of about 10 cm

per year since the installation. Each head is equipped with two light collectors looking upward and downward. The collected

radiation is transmitted by 6-m long fiber optics (core diameter of 400 μm) to the rest of the device that is buried under the snow. Neither electronic nor fragile parts are above-ground.

The instrument scheme is depicted in Fig. 6. The fibers coming from the heads are directly connected to the inputs of a 16-to-1 optical switch (FiberSwitch® mol 1×16 19" 2 HU by Leoni). Using a mechanic-optical module driven by a digital interface, it links in a few milliseconds one of the 16 inputs to the output with a reproducibility better than 0.03 dB according to the manufacturer. This corresponds to about 0.7% uncertainty for the irradiance which in the worse case translates to 1.4% for the albedo. The output of the switch is connected to a splitter 1-to-2 that transmits radiation to two spectrometers (MAYA2000 PRO, Ocean Optics). One covers the visible and near-infrared and the second one is dedicated to the near-infrared. Only the former is used in this study. It gives signal above the noise level in the range 350 - 1050 nm with an effective spectral resolution of 3 nm. One of the inputs of the optical switch is obstructed so it measures dark conditions that are used to correct the offset present in the spectrometer acquisitions. The instrument container is thermalized at +10±2°C which is necessary for nominal operation and stability of the spectrometers.

An embedded computer controls the optical switch, the spectrometers and schedules the acquisition. Every 12 min, a complete sequence of measurements is performed in the following order: head 1 (incident and then reflected channels), head 2 (incident and then reflected channels), buried fibers, heads 1 and 2 again, and at last dark. Each measurement takes 10-20 seconds. An automatic camera is connected to the computer to take one picture per sequence like those shown in Fig. 5. This provides qualitative but very useful information on the state of the snow surface and the presence of frost on the light collectors. The system is connected to the power supply of the Concordia station – the power consumption is typically around 50 W – and to the network to send data in near-real time.

## 3.2 Design and characterization of the light collectors

The light collectors are an important component. Albedo – or more rigorously bi-hemispherical reflectance (Schaepman-Strub et al., 2006) – is the ratio of the upwelling and downwelling spectral irradiances, and the flux is the energy (per unit of surface, of time and of wavelength) crossing an horizontal surface. To measure flux, the collector must have a so-called cosine response so that the energy coming with an oblique direction is weighted by the cosine of the zenith angle (Grenfell et al., 1994; Morrow et al., 2000).

In principle, a flat surface which transmits a constant proportion of the incident beam whatever its direction acts as a perfect collector. For this reason, collectors are often made of a flat disk of highly diffusing material like sintered teflon grains. Nevertheless, teflon grains, like snow grains, have a strong forward scattering behavior so that flat collectors are more reflective at grazing angles than for the normal incidence. As a result, they transmit less at grazing angles than required for a perfect cosine response. In clear sky conditions, when the sun is low on the horizon, the upward-looking collector tends thereby to underestimate the incoming flux. In contrast, the irradiance reflected by the snow is more diffuse and the down-looking collector is almost unaffected by the collector quality. This differential behavior usually results in an over-estimation of the albedo which increases with the solar zenith angle and depends on the direct/diffuse ratio of the sky.

This problem is particularly serious because it strongly depends on the wavelength. This arises because scattering by the light collector materials and the atmosphere strongly decreases with longer wavelengths. Hence, the collector angular response changes with the wavelength from a near-ideal cosine response in the blue part of the solar spectrum to a nadir peaked response at longer wavelengths (Grenfell et al., 1994). This effect is enhanced by the decrease of the direct/diffuse component ratio of the downwelling irradiance under clear-sky conditions, with the wavelength. This results in chromatic aberration that depends on the sun elevation, cloudiness, and other factors affecting the spectral and angular characteristics of the illumination.

To address this difficulty, we pursued a two-fold strategy consisting of building and optimizing our own collectors and applying an *ad hoc* correction to remove the residual imperfections. The latter requires the angular response of the collectors to be measured.

Our collectors are made of a bumped surface cap (Figure 7) following the design proposed by Bernhard and Seckmeyer (1997) to compensate for the reduced transmission at grazing angles. However, a side-effect of this design is that the light transmitted through the cap is not emerging symmetrically with respect to the collector axis (shown in Figure 7) but is more intense in the azimuthal direction of the sun. This effect increases with the solar zenith angle. This has a very negative side-effect because the spectrometer does not collect all the light transmitted by the collector but only a part coming from a rectangular area that corresponds to the optical image of the slit used in input of the spectrometer to limit the aperture before the grating (the fiber does not blur enough this image to weaken this effect). The combination of the eccentered illumination on the cap in the azimuthal direction of the sun and the collection of the light from another unknown azimuthal direction results in a general azimuthal response of the system. This response is about 5-10% in amplitude, which is unacceptable for measuring albedo with 1% accuracy. To solve this critical problem, we inserted a second flat disc between the fiber and the bumped cap whose role is to homogenize the light coming from the cap. Moreover, by adding more scattering material, this improves the overall diffusion of the collector and thereby the quality of the angular response. We could not have used thicker caps or disks to further improve the scattering because the overall transmissivity of the collector is already of the order of 0.001, which imposes integration time of the order of 1 second. Longer integration time would result in increasingly reduced signal-to-noise ratios. The optimization of the shape of the collector cap was done empirically by a series of trials and errors.

To measure the collector response, we set up an optical bench with a collimated light beam illuminating the collector mounted on a motorized rotation stage. The collector is connected to the spectrometer through a flexible fiber optic. Both the spectrometer and the stage are controlled by computer allowing rapid and reproducible measurements of the irradiance $C(\lambda, \theta)$ as a function of the incident angle $\theta$. A typical response is depicted in Figures 8 and 9 as a function of the angle and wavelength respectively. It has been normalized by the ideal cosine and considering exact response at $0°$ following (Grenfell et al., 1994). For angles lower than $70°$, the deviation remains within $\pm 6\%$ of the ideal response which is comparable to Grenfell et al. (1994) but twice better than Carmagnola et al. (2013) who used a flat collector. The deviation between $70°$ and $80°$ degrades significantly which suggests uncorrectable measurements in this range of SZA, however it remains within $\pm 15\%$ while Grenfell et al. (1994) show twice larger errors. The values under 500 nm are extrapolated (Fig. 9) because our light source is too weak in this wavelength range to get signal above noise level. The response is relatively independent of the

wavelength for SZA=50° and becomes more and more wavelength-dependent at larger angles. The measured responses are used for the *ad hoc* correction described in the next section.

## 3.3 Processing of raw measurements into albedo

The calibration of the raw spectra acquired by the spectrometer requires several processing steps: 1) dark and stray light corrections (abbreviated 'dc' and 'sl' hereinafter), 2) integration time scaling ('it'), 3) cross and absolute calibrations ('cal'), 4) cosine correction and cloud detection ('cc'), 5) albedo calculation. These steps are described in details in the following:

**Dark and stray light corrections**. The raw spectra acquired by the spectrometer are in numerical counts ranging from 0 to about 65000 for each pixel of the CMOS sensor (16-bit Analog-to-Digital converter). As any photo-sensitive sensor, the spectrometer is subject to dark current and other effects that result in non-zero outputs even when no light enters in the device. It is important to remove this offset as highlighted in Section 2.3.1 especially in weak illumination conditions or near the borders of the bandwidth where the sensor sensitivity is the weakest. The dark current of our spectrometers depends on temperature and integration time and may be subject to aging of the spectrometer and other environmental parameters. Because it is not possible to control all these parameters, dark spectra are measured in every sequence (every 12 min) using the dedicated channel. Since we found a linear relationship between the offset and the integration time, acquisitions at two extreme integration times (T1=13 ms the minimum of the spectrometer and T2=1000 ms the typical time for our measurements) are necessary and sufficient to correct all the spectra acquired during the same sequence at any integration time. Hence, to correct a spectrum $S^T(\lambda)$ acquired with a given integration time $T$, we first estimate the offset at the actual acquisition time by linear interpolation of the two extreme dark spectra ($D^{T1}(\lambda)$ and $D^{T2}(\lambda)$) and then subtract this estimate from the measured spectrum as follows:

$$S^T_{dc}(\lambda) = S^T(\lambda) - \left[ \frac{D^{T2}(\lambda) - D^{T1}(\lambda, T_1)}{T_2 - T_1}(T - T_1) + D^{T1}(\lambda, T_1) \right] \tag{12}$$

where $S^T_{dc}(\lambda)$ is the dark current corrected spectrum.

Stray light is another typical artifact of spectrometers (Kantzas et al., 2009). It comes from the small fraction of the beam that instead of passing through the grating to be dispersed is deviated and after internal reflections ends up on the CMOS sensor adding signal to the pixels. Second order dispersion from the grating is another cause of stray light but our spectrometer is mounted with a second-order filter that prevents this effect. For spectrometers of the same brand as ours, Kantzas et al. (2009) found stray light amounting to between 2 and 12% of the maximum intensity. This effect results in an offset which depends on the total intensity entering the spectrometer. To estimate the stray light, we assume it affects all the pixels equally (i.e. independent of the wavelengths) and estimate it using the average of the counts in the range 200–260 nm where no radiation is supposed to come from the grating, because the fiber optics cut-off is around 290 nm and because the weak sensitivity of CMOS sensor in this range. The spectrum corrected from stray light writes:

$$S^T_{sl+dc}(\lambda) = S^T_{dc}(\lambda) - \left\langle S^T_{dc}(\lambda) \right\rangle_{\lambda=200-260nm} \tag{13}$$

**Integration time scaling**. Considering that the CMOS sensor accumulates charge with the same efficiency whatever the integration time, the spectra in counts are divided by the integration time $T$ to obtain a signal proportional to the incoming radiance:

$$S_{it+sl+dc}(\lambda) = S_{sl+dc}^{T}(\lambda)/T \tag{14}$$

**Calibrations**. The calibration of each channel to obtain absolute irradiance is performed in two steps, first the cross-calibration of the channels to each others and second the absolute calibration of the set. We came to this two-step strategy because it was difficult to design an experiment to perform the absolute calibration directly with a reproducibility better than 1% as required for our target accuracy. Since the albedo only depends on the relative calibration between the channels, we put more effort on the cross-calibration than on the absolute one.

The experiment to obtain data for the cross-calibration consists in measuring successively the upward and downward channels under the same illumination conditions. To do this, before the installation, we first acquired spectra with the head in normal position, with special care at the horizontal leveling, and performed a second acquisition immediately after flipping the head up side down. The elapsed time between these operations was about 30 s and the experiment was conducted under clear-sky conditions which ensured constant illumination. For each channel, the spectrum taken when looking downward $S_{it+sl+dc}^{\text{cross}}(\lambda)$

is used to calibrate any spectra acquired after installation for this particular channel (whatever the direction of looking once installed) as follows:

$$S_{ccal+it+sl+dc}(\lambda) = \frac{S_{it+sl+dc}(\lambda)}{S_{it+sl+dc}^{\text{cross}}(\lambda)} \tag{15}$$

where the *ccal* subscript refers to the cross-calibration step. The downward direction is chosen instead of the upward because the illumination coming from the snow is mostly diffuse which limits the sensitivity to the head leveling. The experiment was

done just before the installation independently for both heads so that it is useful to cross-calibrate both channels of the same head (which is relevant for the albedo) but is inadequate for inter-header calibration.

The absolute calibration aims at inter-calibrating the heads and provide absolute irradiance (in $\text{W m}^{-2}\,\text{nm}^{-1}$ for instance). The experiment to collect reference spectra was done after the installation of Autosolexs and uses a commercial spectral radiometer (HR1024, Spectra Vista Corporation) as the reference. Spectra have been acquired simultaneously with the two

instruments and for the two heads which allows any spectra acquired later from any channel to be normalized as follows:

$$S_{cal+it+sl+dc}(\lambda) = S_{ccal+it+sl+dc}(\lambda)\frac{\text{SVC}(\lambda)}{S_{ccal+it+sl+dc}^{\text{abs}}(\lambda)} \tag{16}$$

where $\text{SVC}(\lambda)$ and $S_{ccal+it+sl+dc}^{\text{abs}}$ are the spectra taken with the reference spectral radiometer and Autosolexs respectively. We do not expect an accuracy better than 20% for this step owing to the limited intrinsic accuracy of the reference and the many possible artifacts of the experiment. However this does not impact the albedo as explained before.

**Collector angular response correction**. The imperfect angular response of the collector mainly affects the incident channel under clear-sky conditions. Following Grenfell et al. (1994), we use the collector response measured in the laboratory $C_{it+sl+dc}(\lambda,\theta)$ at different angles $\theta$ and corrected for the dark current, stray light and integration time beforehand. Azimuthal

symmetry was checked and appeared to be very good. The correction is applied only to the direct component as the diffuse component has already been calibrated at the absolute calibration step, hence only the incident channel is concerned:

$$S_{cc+cal+it+sl+dc}(\lambda) =$$

$$S_{cal+it+sl+dc}(\lambda) \left[ r^{\mathrm{diff}}(\lambda,\theta) + \left(1 - r^{\mathrm{diff}}(\lambda,\theta)\right) \frac{2\cos\theta}{C_{it+sl+dc}(\lambda,\theta)} \int\limits_0^{\pi/2} C_{it+sl+dc}(\lambda,\theta') \sin\theta' d\theta' \right] \tag{17}$$

where $r^{\mathrm{diff}}(\lambda,\theta)$ is the diffuse fraction. The first term in the brackets on the right-hand side represents the diffuse contribution which is assumed isotropic. The anisotropy of the sky diffuse component is neglected which is second order as long as the correction is small. The second term is the direct contribution that is normalized by the collector angular response ($C_{it+sl+dc}(\lambda,\theta)$) relatively to the ideal cosine ($\cos\theta$).

The $r^{\mathrm{diff}}(\lambda,\theta)$ quantity is estimated using the SBDART atmospheric model (Ricchiazzi et al., 1998) considering clear-sky conditions. The atmosphere for summer Arctic is used with the altitude adapted to Dome C. The column water vapor totals 0.4 mm (Tremblin et al., 2011) and ozone is set to 300 DU. Aerosol optical depth at 440 nm is set to 0.02. This calculation does not account for the temporal variations of the atmospheric composition. It could be improved in the future by exploiting the measured incident spectra to infer and better represent the actual atmospheric conditions.

For the present study, we filtered out the data acquired in cloudy conditions. To detect such conditions, the incident spectrum is compared to a spectrum chosen as a reference when the sky was known to be clear (2013-11-15 03:36 UTC close to the local zenith). Each spectrum is then scaled beforehand by the cosine of the solar zenith angle to account for the geometrical difference. When the scaled spectrum is more than 10% lower than the reference spectrum in the range $405 - 550$nm, we consider the sun as being partially obstructed and the data are discarded.

Eventually the spectral albedo is calculated by taking the ratio between the upwelling and downwelling spectra. The shadow of the mat and the heads on the snow surface is very small compared to when an operator takes manual measurements (Carmagnola et al., 2013) and is neglected.

Before applying the SSA retrieval algorithm, albedo spectra are smoothed to reduce the noise using a first order Butterworth low-pass provided by the Python scipy.signal.butter function with cut-off of 0.1.

### 3.4 Illustration of the processing steps

The effect of the processing steps is illustrated in this section for two acquisitions taken in contrasting solar conditions: a first one at local noon the 10 January 2013 with the sun relatively high in the sky for Dome C (SZA = 53°) and a second one at 8 pm the same day (SZA = 77°). Figures 10 (a) and (b) illustrate the contribution of each step (curves with different colors) for these acquisitions.

The first step (dark correction) is shown in Figure 10 (orange). It removes the overall offset that is clearly visible in the raw data at wavelengths where the CMOS sensor has little sensitivity (under 350 nm and around 1100 nm). This correction has a positive side-effect as it also reduces the small peaks due to damaged pixels like at 862 and 1069 nm. With the aging of the spectrometer, the number of such pixels tended to increase (data not shown), but they remain isolated and the correction

is efficient. The albedo is affected by this correction especially near the margins of the spectrum where the sensitivity of the CMOS sensor fades. Because both the incident and reflected spectra are equally affected by the same offset value, the albedo calculated with un-corrected data tends to 1 near the margins. Since clean snow albedo is expected to tend to about 0.98 – 0.99 in the green and blue wavelengths, un-corrected data may appear better, but it is an artifact and the offset really needs to be corrected. At the other side of the spectrum, in the near-infrared, the correction significantly changes the shape of the absorption feature around 1030 nm by lowering the local minimum which has a huge impact on the SSA estimation. Applying the SSA retrieval algorithm on the albedo spectra , we find that SSA drops from 53 to 41 $\mathrm{m^2\,kg^{-1}}$ after the correction for the noon acquisition and from 61 to 24 $\mathrm{m^2\,kg^{-1}}$ for the evening one. These huge differences highlight the necessity of this correction.

To evaluate this processing step within the modeling framework developed in Section 2.3.1, we consider typical values of the noon acquisition: the spectrum has an amplitude of nearly 50000 and the dark current (offset) is around 3800. We further assume that this offset can estimated with an accuracy of 10% (pessimistic estimate for illustration), which results in $d' = 0.8\%$. Figure 11 shows the perfect albedo (black curve) calculated for a true SSA of 50 $\mathrm{m^2\,kg^{-1}}$ and the perturbed albedo (green curve) with $d' = +0.8\%$. Despite the small difference between the spectra, the estimated SSA is 55.4 $\mathrm{m^2\,kg^{-1}}$ which corresponds to a relatively large over-estimation of 10%, but remains under our criteria of 15%. Considering that the dark current can be usually estimated by much better than 10%, the impact on the SSA is always minor.

The next processing step (stray light correction) is not shown because it has a negligible effect. Nevertheless, it is worth noting that we assume the stray light effect to be uniform on all the pixels of the CMOS sensor. If this is not the case, the correction of the stray light would require a more complex method and the evaluation of the error proposed here would be only a lower bound of the error.

The cross and absolute calibration steps (from orange in top graph to violet curve in middle graph) also radically change the shape of the spectra. The most visible change on the radiance is mainly due to the absolute calibration because it takes into account the sensitivity of the spectrometer. As expected for a silicon-based sensor, the correction increases the signal in the blue and near-infrared relatively to the yellow and red. Nevertheless, the absolute calibration is not important for the albedo as explained before. The difference on the albedo graphs before (orange curve) and after (violet curve) the calibrations is only due to the cross calibration step between the downward- and upward-looking channels and corresponds to a scaling that linearly decreases from 0.95 in the blue to 0.85 in the near-infrared. Because of this wavelength dependence, the SSA estimation is strongly affected, values decrease from 41 $\mathrm{m^2\,kg^{-1}}$ before correction to 36 $\mathrm{m^2\,kg^{-1}}$ after for the noon acquisition and from 24 $\mathrm{m^2\,kg^{-1}}$ to 20 $\mathrm{m^2\,kg^{-1}}$ for the evening one. In the theory presented in Section 2.3.2 this corresponds to approximately $b = 0.10$. Using Figure 4, it means that the cross-calibration is definitely required to reach the 15% accuracy of SSA. If we assume that the correction is precise within 10% (for instance because of the limited reproducibility of the experiment to determine the reference spectra) it leads to a residual trend of $b \approx 0.01$ which is weak enough to reach the 15% accuracy. In addition, it is worth noting that this correction is constant over the whole time-series and has therefore no impact on the relative temporal variations of SSA.

The collector angular response correction (from violet to pink curves) has very small impact on the radiance (middle graph) whatever the acquisition hour. The impact is also apparently moderate on the albedo (bottom graph). The correction factor ranges from 0.98 in the blue to 0.99 in the near infra-red at noon and 1.00 to 1.07 at 8 pm. Even if these values are weak (maximum of 7%), the correction succesfully removes the decreasing trend in the shorter wavelengths of the visible (400 to 600 nm) that affects the evening acquisition. Albedo measurements by Nicolaus et al. (2010) show a similar artifact. Such a trend is visible throughout the timeseries at large SZA and the collector response correction usually performs well by recovering a nearly constant value as expected.

Regarding SSA estimation, the correction has a little impact for the noon acquisition (36.7 and 36.2 $\mathrm{m^2\,kg^{-1}}$ respectively before and after the correction) as expected. In contrast, for the evening acquisition the SSA estimate increases from 20 $\mathrm{m^2\,kg^{-1}}$ to 35 $\mathrm{m^2\,kg^{-1}}$, the latter being close to the value at noon. This clearly shows that the correction of the collector response is crucial. The theoretical spectrum that fits the fully-corrected spectrum is shown in black in Figure 10. The differences between the fit and the observation are small, we can note a slight over-estimation at 800 nm and under-estimation at 970 nm which are very likely due to a small error in the refractive index of the ice already pointed out in other studies (e.g. Carmagnola et al., 2013). In contrast, the over-estimation at 1030 nm is more likely an error in the observations resulting from the low sensitivity of the spectrometer in this wavelength range.

## 3.5 Stability

Before deriving albedo and SSA over long periods of time, it is important to assess the overall stability of the instrument.

The leveling is a first and important parameter. It was carefully done at the installation of Autosolexs in December 2012 within the uncertainty allowed by spirit level (about $0.2°$) but could have degraded. Nevertheless, in December 2015, new measurements with an electronic devices yielded insignificant movement of the structure.

The radiometric stability is another parameter. The seasonal variations of downwelling irradiance are plotted in Figure 12 for the two heads and for the three summer seasons. The irradiance is integrated between 400 and 1000 nm and averaged between 10h – 14h (local time). These data include all weather conditions. Overall, there is a good year to year agreement. For instance, the irradiance observed by the head 1 during the 30 days after the solstice (period in common for the three seasons) has increased by +0.5 % and +4.2% the second and third years with respect to the first one. These values become -0.5%, and +3.0% respectively for the head 2, which is relatively small for an unattended instrument and considering that these values include the inter-annual variability of cloudiness. Only the third season shows a significant increase (around 4-5 %) which is unexplained.

## 4  Observations, interpretation and comparison

This section presents the 3-year long time-series of retrieved SSA and provides an analysis of its quality and information content.

## 4.1 Diurnal cycle of SSA

The SSA estimated over the course of a day for each head is depicted in Figure 13a for the 10 January 2013 as a function of the local hour and the corresponding SZA. Gray symbols represent data rejected for one of the three reasons: presence of clouds, excessive chromatic aberration or $A$ outside the range $0.9 - 1.1$.

This particular day clouds were detected only at 3:25 and after 21:00 but the detection may not be reliable at high SZA because of the simplistic approach used to scale the reference spectrum to a given SZA (Section 3.3). The other cases of rejection are equally due to the two other criteria. It appears clearly that it usually happens when the sun is low on the horizon, with a SZA higher than $70°$, which is expected. However, in many cases, the SSA of rejected acquisitions is close to valid ones which could indicate that our criteria are too conservative. On the other hand, some data at very high SZA are not rejected while we believe they should have been. For this reason, we let the criteria being conservative and even discarded acquisitions at SZA higher than $75^o$ in the following.

Valid data show small diurnal variations between 32 and 35 $\mathrm{m}^2\,\mathrm{kg}^{-1}$ for head 1 and slightly larger between 31 and 40 $\mathrm{m}^2\,\mathrm{kg}^{-1}$ for head 2. Root mean squared variations is respectively 0.9 and 2.6 $\mathrm{m}^2\,\mathrm{kg}^{-1}$ which is well under the 15% target accuracy. The mean difference between both heads is 2.6 $\mathrm{m}^2\,\mathrm{kg}^{-1}$. The nearly perfect symmetry around noon suggests that these variations are not real but are caused by SZA-dependent errors either due to uncorrected artifacts in the observations or limitation of the theoretical model used for the estimation.

The diurnal cycle is shown in Figure 13b (1 March 2013). The SSA is much higher than in the previous case and the number of hours with SZA lower than $75°$ is more limited. In addition, more data are rejected, mostly because of the chromatic aberration. Altogether, the valid values range between 53 and 74 $\mathrm{m}^2\,\mathrm{kg}^{-1}$ representing a large amplitude of 30%. The standard deviation is about 3.5 $\mathrm{m}^2\,\mathrm{kg}^{-1}$ for both heads, that is 10% for a $\pm1\sigma$ variation. It was impossible to determine whether these variations are real or result from measurement and retrieval errors. These variations must therefore be interpreted as an upper bound estimate of the uncertainty of the retrieved SSA.

## 4.2 Statistical characteristics of the seasonal variations

The seasonal variations of SSA estimated from the two heads over three summers (December 2012 to March 2015) are shown in Figure 14. The daily mean of valid data is plotted in color and each individual valid data is shown in blue in the background to give an idea of the diurnal variability (or maybe residual error). In this paper, we perform a quality assessment of this series. The interpretation of the geophysical features is addressed in Libois et al. (2015).

At the beginning and end of the seasons, the SSA is usually high and exhibits large day to day variations that are suspicious. Several issues like the those due to high SZA, the weaker sensitivity of the SSA – albedo relationship at higher SSA (Domine et al., 2006; Gallet et al., 2009) and the lower number of valid data entering the daily mean during these periods with short daylight, probably combine during these periods. Because of this lower quality, the time-series in Libois et al. (2015) only uses noon data to minimize the high SZA issue and is limited to SZA lower than $67°$ which corresponds at Dome C to the period 18 October to 27 February. Here, because we introduce the filter based on the detection of chromatic aberration, we apply the

algorithm for SZA up to 75°, relying on the filter to reject low quality data at high SZA. As a result, the period presented is slightly longer (about two weeks each sides but with little data), but the interpretation at the extremes is still subject to caution. Using the daily mean of the valid data instead of the noon value has very little impact, the difference ranges between -1.0 and +1.5 $m^2\,kg^{-1}$ depending on the season and head.

5    The two heads give very close values (within the daily range of variation) for some periods while significant differences are observed for others usually in the spring (October 2013 and most remarkably October-November 2104) and in the fall (March 2013 and 2014). If we consider December-January to assess the statistics, we find the difference between heads 1 and 2 to be -2.7, 1.0 and 0.0 $m^2\,kg^{-1}$ on average and 3.2, 2.4 and 2.2 $m^2\,kg^{-1}$ root mean square (rms), respectively for the three seasons. This rules out any significant bias between the two heads which could have come from issues with the cross-calibration, 10  the only processing step that uses different data depending to the head. This also indicates that the seasonal stability of the instrument and processing is of the same order as the diurnal standard deviation (Section 4.1). Conversely, it means that the difference observed during some periods might originate from the difference of SSA between the footprints of the two heads (Section 4.3).

    Altogether these results converge to a precision much better than the targeted 15% which ensures that the seasonal variations 15  can be interpreted as a geophysical signal (Libois et al., 2015). However, it does not presume of the absolute accuracy which mostly depends on the validity of the model in general and the choice of the parameter values (shape factors, assumption of a flat surface...).

### 4.3   Horizontal variability between both heads

The two heads located 2-m apart show very similar variations most of the time (Figure 14) and agree on the overall seasonal 20  decrease from values higher than 60–80 $m^2\,kg^{-1}$ in October to a minimum around 25 $m^2\,kg^{-1}$ occurring in January and/or February. However, they also significant differ during some periods. Being at 2 m above the surface and accounting for the cosine weighting of the collector, we can estimate that 50% of the received signal comes from a disk of $2\,m$ in radius ($12\,m^2$ in surface area), and 75% from a disf of $3.5\,m$ in radius ($38\,m^2$). Despite the proximity and a slight partial overlap of the footprints, differences between the two heads can be particularly marked as in Spring 2014. We could invoke the fact that 25  snow accumulation is irregular at Dome C (Petit et al., 1982; Libois et al., 2014a) leading to irregular deposition on the surface. However, this argument is not compatible with the absence of difference observed during the summers although several precipitation events usually occurred with a clear effect on the observed SSA (Libois et al., 2015) but did not produces differences between both heads.

    In spring 2014, the large and persistent difference could be ascribed to the presence of an erosional feature under head 1 30  (Figure 5) and a wind-form feature (elongated dune) under head 2. The latter could be composed of recently deposited snow while the former feature sastrugi and older snow layers emerging on the surface. Nevertheless, this interpretation is subject to caution because visible photography is insensitive to SSA, as evidenced by the resemblance of the two pictures taken 12 October and 27 November despite a three-fold decrease of the SSA.

During this period, the diurnal variations of SSA are not symmetrical with respect to noon (not shown) as opposed to the cycles shown in Figure 13. A possible reason is the inhomogeneity of the surface SSA. Since the surface is particularly rough and the sun low on the horizon during this period, the shadows are significant and change with the solar time. Hence, the "average" spectrum measured by the instrument comes from different areas depending on the time of the day and those areas may have different SSA. The direct effect of the roughness could also affect the symmetry of the diurnal cycle when the surface is rough but assessing the spectral effect of the roughness is difficult and beyond the scope of this study.

## 4.4 Comparison with ASSSAP

Manual measurements of surface snow SSA were taken with ASSSAP (Arnaud et al., 2011; Libois et al., 2015) a device using reflectance at 1310 nm to estimate SSA using similar theory as used here. For these measurements, we followed a specific protocol that takes care to preserve the surface and prevent post-sampling transformation of the snow. More than 5 manual measurements were taken every day in a site located approximately 1 km from Autosolexs. The sampling strategy was not random but aimed at covering the variety of different surface facies identified by the operator. It implies that the average of the values may differ from the true areal average of SSA when the spread of SSA is large.

Figure 15 shows individual manual measurements taken during the 2013-2014 seasons with ASSSAP along with Autosolexs SSA time-series (average of the two heads). Figure 15 clearly shows that most manual measurements significantly differ from the SSA estimated from Autosolexs. However the lowest values seem to agree with Autosolexs during the first part of the season, until about 8 January. After this, the spread of manual measurements is significantly reduced and despite higher values of SSA with ASSSAP, the temporal variations are correlated with those of Autosolexs.

This comparison clearly demonstrates that the SSA retrieved in different wavelength ranges differ, which we attribute to strong heterogeneity of SSA between the uppermost 5 mm and 1 cm according to the calculation presented in Section 2.2.

The large difference in the first part of the season can be attributed to the frequent blowing snow events occurring during this period but the difference persists over 8 January despite calmer conditions. This persistence of large difference of SSA over a small vertical distance is not well understood (Gallet et al., 2011). Clear-sky precipitation (a.k.a diamond dust) is a possible candidate but snow evolution simulation performed by Libois et al. (2015) uses ERA-Interim as input. The reanalysis successfully forecasts the precipitation events which leads to marked rises of SSA (Figure 14) but does not include diamond dust. Since the simulation agrees with the SSA observations, this indicates that diamond dust is not required in the model to get a sustained high surface SSA on the surface. We can conclude that a significant SSA gradient in the first uppermost centimeters exists with SSA possibly exceeding $100 \, \mathrm{m^2 \, kg^{-1}}$ on the very surface and that this gradient persists over long periods, even during periods without precipitation and blowing snow.

## 5 Discussions and Conclusion

This study introduces an algorithm to retrieve the specific surface area of the surperficial snow from spectral albedo measurements, along with Autosolexs, a multi-channel spectral radiometer used to automatically measure spectral albedo in the visible

and near infrared range (under 1000 nm). Despite the specificity of this home-made instrument and the operating conditions of Dome C, several general recommendations can be made regarding the design and deployment of such devices: 1) Using fiber optics to transmit the light makes possible to bury the temperature-sensitive components of the instrument under the snow. This was necessary for Dome C considering that temperature is frequently below -40°C even during the daylight. Maintaining the

device at constant above freezing temperature avoids deterioration and drift of the spectrometer. However, it costs some energy which would require adequate dimensioning of solar panels or wind turbines in the case of autonomous stations. The instrument has been working since December 2012, even in winter, with only a few short interruptions due to power outage. 2) Use of optical switch instead of two or more spectrometers is a good option. It is cost effective, avoids the need of inter-calibration of the spectrometers (Nicolaus et al., 2010) which may drift independently over time (France et al., 2011) but it is subject to

switching delay and reproducibility, and involves mechanical parts. It allowed us to deploy 2 measurement heads located a few meters apart which provided evidence of spatial difference during some periods. With a 16-channel optical switch, it is technically possible to set up 7 measurement heads 3) The leveling of the head should be and remain horizontal not only for albedo which has been highlighted by numerous authors (e.g. Bogren et al., 2016), but also for SSA retrieval. Under typical polar summer conditions or in winter in the mid-latitudes, we estimate the leveling must be better than 1° to get at least the

noon acquisition accurate enough to retrieve estimate SSA with an accuracy of 15%. Maintaining automatically the leveling and measuring the inclination at low temperature is challenging. 4) Designing good light collectors and correcting for their imperfect response is necessary for SZA above 60° and becomes more and more critical above 70°. The determination of the atmospheric diffuse/direct ratio and cloud mask deserves further work compared to what has been implemented here. Using the measured incident spectrum to infer the actual ratio is an avenue. 5) Filtering data based on the difference between observed

and estimated albedo in the visible range discards data subject to any artifact that produces excessive chromatic aberration. The approach developed here does not require high spectral resolution data, it can be applied to multi-band radiometers like those carried by satellites. However, it requires some assumptions like negligible impurity content and could interfere with the atmospheric correction often based on the blue channels for space-borne sensors.

Part of the difficulties addressed in this study arises from the weak sensitivity of the SSA-reflectance relationship at wave-

lengths under 1 μm, especially for the high SSA encountered at Dome C ($50 \, \mathrm{m^2 \, kg^{-1}}$ and higher) (Domine et al., 2006; Gallet et al., 2009). Extending the present instrument further in the infrared seems attractive because the sensitivity to SSA is greater so that larger error on the albedo measurement would be acceptable for a similar target accuracy of SSA. However it comes at a price. Building light collectors with a good angular response is increasingly challenging at longer wavelengths due to the decreasing scattering. The balance between theoretically more sensitive wavelengths and increasing chromatic aberrations

needs to be weighted.

The influence of the surface roughness on the albedo and estimated SSA also deserves further work. Warren et al. (1998) explained that "introducing roughness will cause no change to the albedo of a surface whose albedo is 0.0 or 1.0. The greatest effect is for intermediate values of albedo; in the case of snow, this means near-infrared wavelength" which implies that roughness change tends to reduce albedo in the 1030 nm absorption feature more than at shorter wavelengths. This implies that

our algorithm, by assuming a flat interface, would under-estimate the SSA over rough surfaces. Moreover, with measurements

heads only 2 m above the surface, the scale of the measurements and that of the roughnesses are of the same order resulting in a more complex problem than that considered in modeling studies (Leroux and Fily, 1998). Spatial survey of albedo and SSA variations could be used to evaluate this effect. The anisotropy of the roughness and the slope of the terrain (relevant to mountainous regions only) are expected to introduce a spurious diurnal cycle and need to be considered in the future as well.

5      The present study, by retrieving 3-year long time series of SSA, confirms the general coarsening of snow grains during the summer observed in several studies (Jin et al., 2008; Picard et al., 2012) and recently shown to be predicable by snow metamorphism modeling (Libois et al., 2015). However, despite its length, it features relatively little inter-annual variations compared to that observed by satellite and provides little evidence yet of modulation of the grain size by precipitation at the seasonal timescale as hypothesized by Picard et al. (2012). An important perspective of this work is to maintain Autosolexs at

10    Dome C and to develop the installation of ground-based spectral radiometers in Antarctica.

*Acknowledgements.* This study was supported by the ANR program 1-JS56-005-01 MONISNOW. The authors acknowledge the French Polar Institute (IPEV) for the financial and logistic support at Concordia station in Antarctica through the NIVO program. This work has also been supported by a grant from OSUG@2020 (investissement d'avenir – ANR10 LABX56). We thank the IPEV winter-over staff at Concordia for monitoring our snow instruments, including Autosolexs.

15    *Author contributions.* G. Picard and L. Arnaud developed and built Autosolexs, L. Arnaud and Q. Libois deployed it at Dome C and performed the calibration experiments. G. Verin performed the collector characterization. M. Dumont conducted the atmospheric simulations. G. Picard and Q. Libois developed the retrieval algorithm and prepared the manuscript with contributions from the other authors.

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

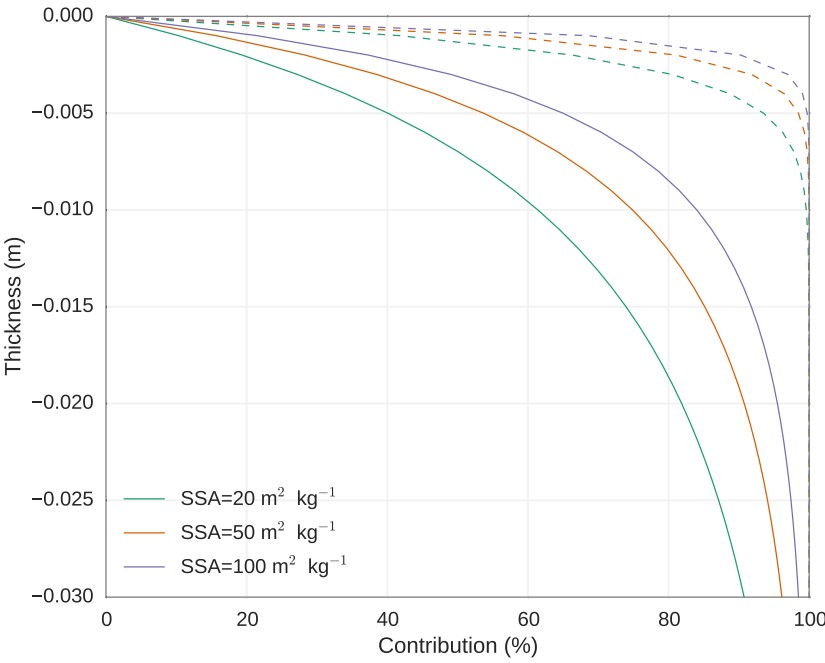

**Figure 1.** Contribution of the uppermost layer of a semi-infinite snowpack to the albedo averaged over the range 700–1050 nm (plain line) and at 1310 nm (dash line) as a function of the layer thickness.

Warren, S. G. and Wiscombe, W. J.: A Model for the Spectral Albedo of Snow. II: Snow Containing Atmospheric Aerosols, Journal of Atmospheric Sciences, 37, 2734–2745, doi:10.1175/1520-0469(1980)037, 1980.

Warren, S. G., Brandt, R. E., and O'Rawe Hinton, P.: Effect of surface roughness on bidirectional reflectance of Antarctic snow, Journal of Geophysical Research, 103, 25 789, doi:10.1029/98je01898, http://dx.doi.org/10.1029/98JE01898, 1998.

5  Warren, S. G., Brandt, R. E., and Grenfell, T. C.: Visible and near-ultraviolet absorption spectrum of ice from transmission of solar radiation into snow, Appl. Optic., 45, 5320–5334, 2006.

Wuttke, S., Seckmeyer, G., and König-Langlo, G.: Measurements of spectral snow albedo at Neumayer, Antarctica, Annales Geophysicae, 24, 7–21, doi:10.5194/angeo-24-7-2006, http://dx.doi.org/10.5194/angeo-24-7-2006, 2006.

Zhuravleva, T. B. and Kokhanovsky, A. A.: Influence of surface roughness on the reflective properties of snow, Journal of Quantitative Spec-
10  troscopy and Radiative Transfer, 112, 1353–1368, doi:10.1016/j.jqsrt.2011.01.004, http://dx.doi.org/10.1016/j.jqsrt.2011.01.004, 2011.

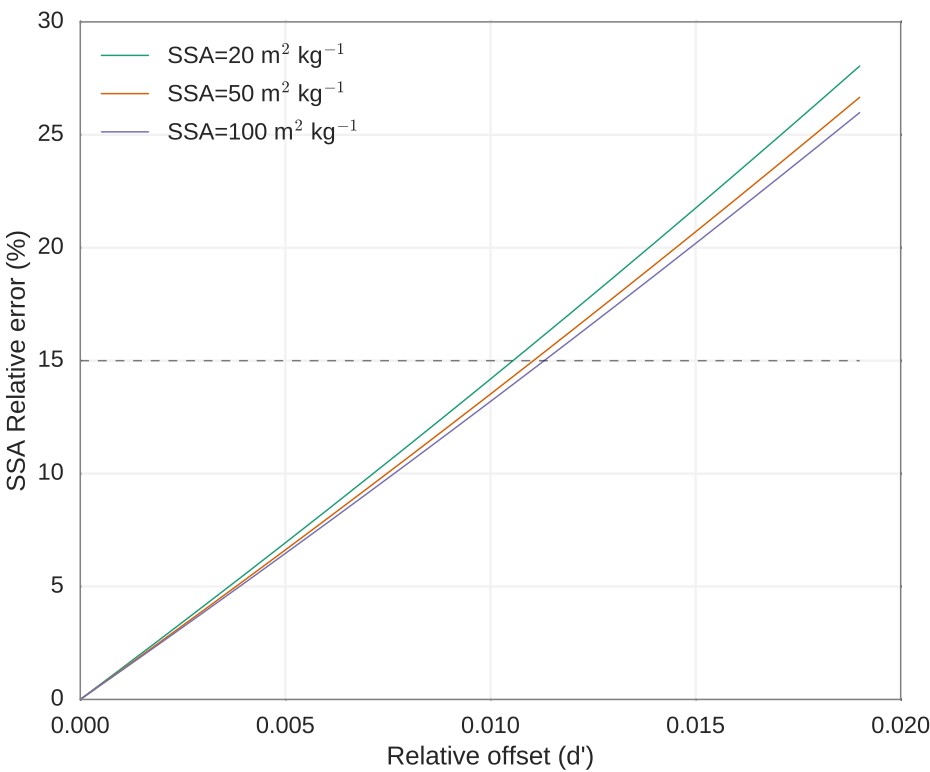

**Figure 2.** Theoretical calculation of the relative error in SSA due to an offset in the incident and reflectance spectra for SZA=60°. The dashed line shows the 15% target accuracy discussed in the paper.

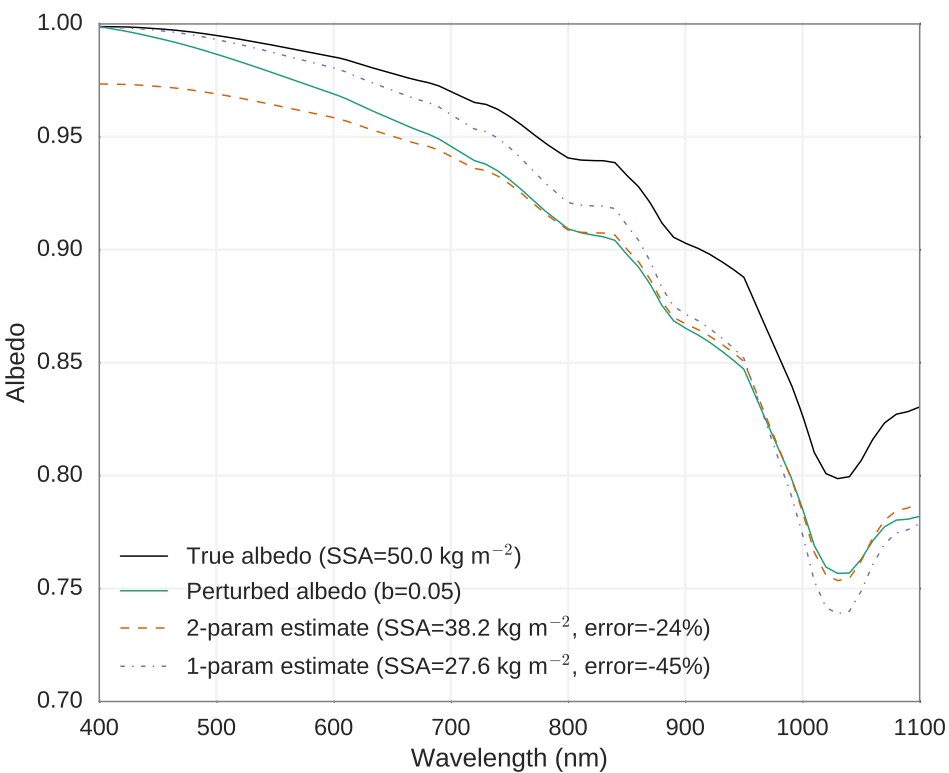

**Figure 3.** Simulations of albedo spectrum for SZA=60°. A perfect albedo spectrum for an homogeneous snowpack with SSA of $50\,\mathrm{m^2\,kg^{-1}}$ (black) is perturbed by a factor linearly decreasing from 1 at 400 nm to $1-b$ at 1100 nm with $b=5\%$ (green) and used as an observation to estimate SSA with the 1-parameter (violet thin dash) and 2-parameter models (orange, long dash). Estimated values are shown in the legend along with the relative error.

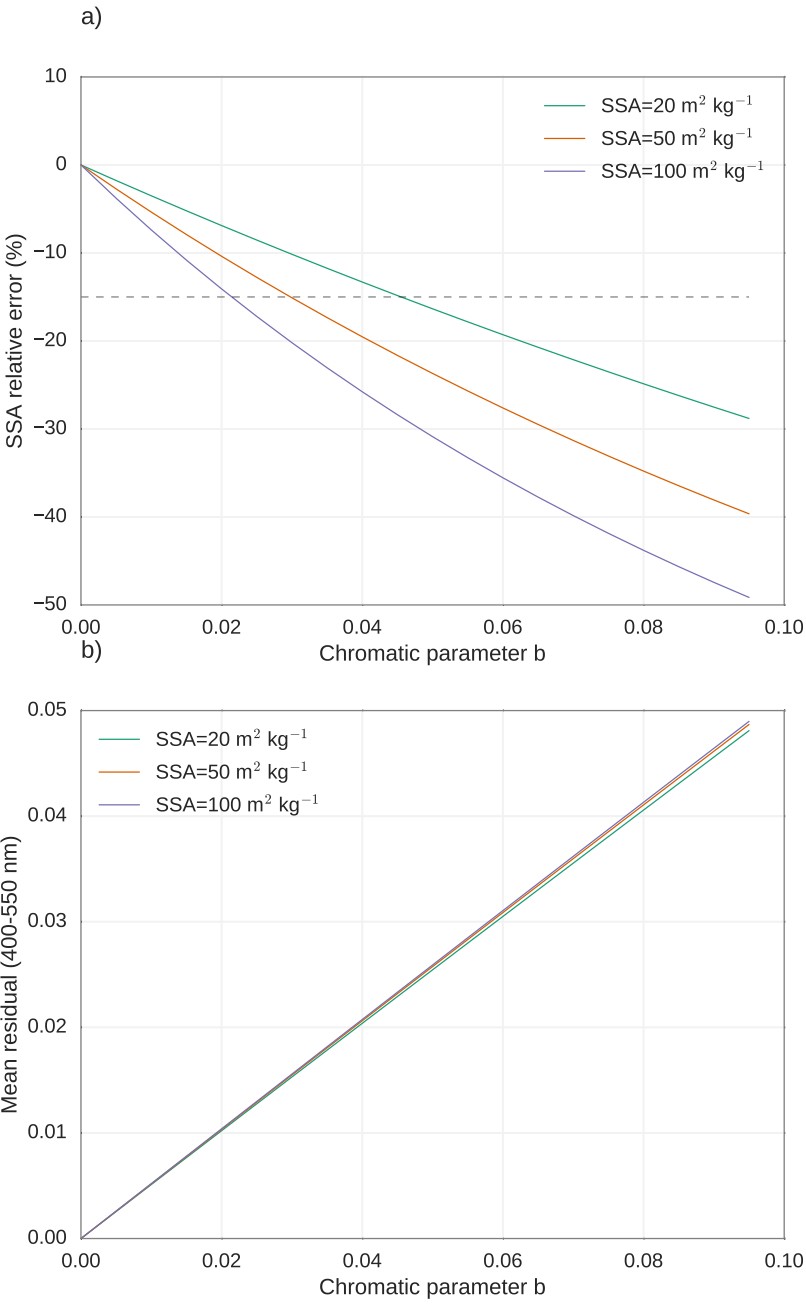

**Figure 4.** (a) Theoretical calculation of the relative error in SSA due to the chromatic aberration of the spectrometer as a function of the parameter $b$ defined in Equation 8 for SZA=60°. The dashed line shows the 15% target accuracy discussed in the paper. (b) Relationship between $b$ and the residual between the observed and estimated albedo averaged over the range 400–550 nm.

12 October 2014

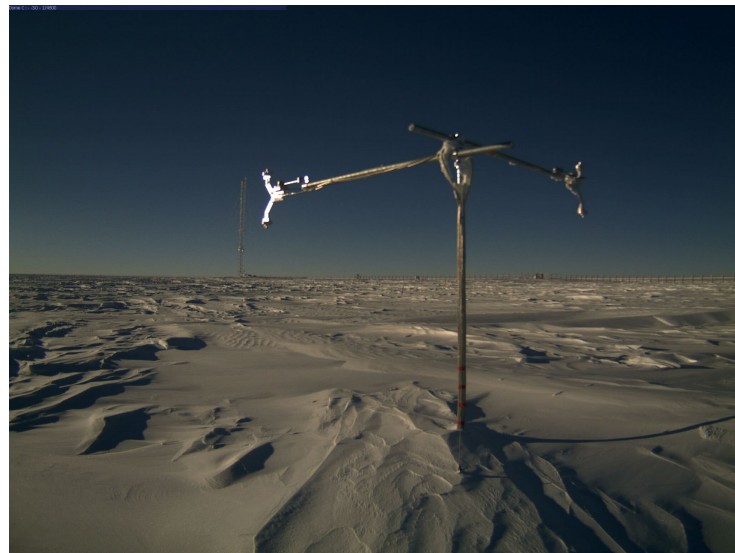

11 November 2014

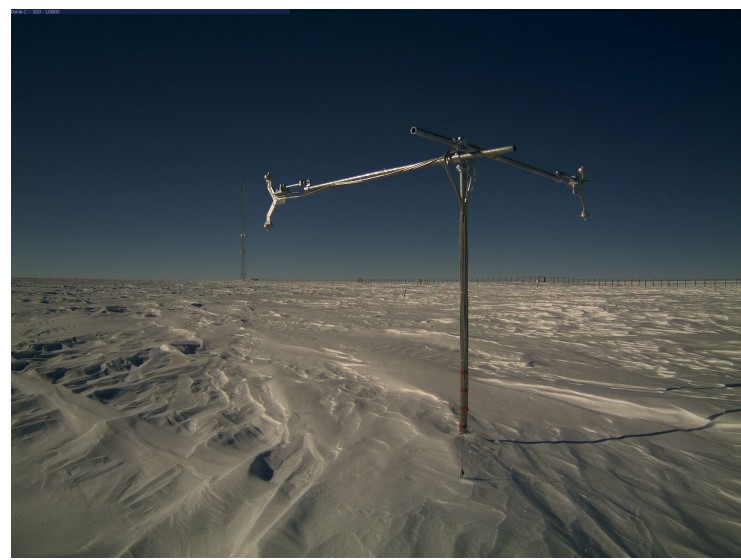

**Figure 5.** Pictures of the snow surface under Autosolexs head (head 1 is on the left) on 12 October 2014 and 11 November 2014.

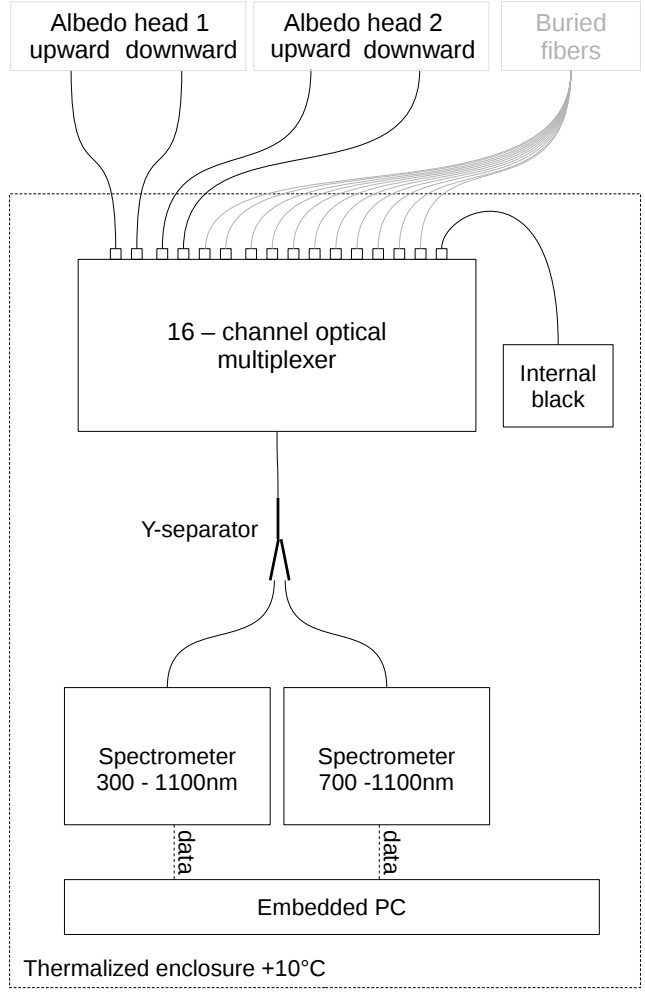

**Figure 6.** Principle of Autosolexs. The dotted rectangle represents the thermalized container that is buried in the snow. The plain curved lines represent fiber optics, the fibers buried in snow (gray) are not used in the present study.

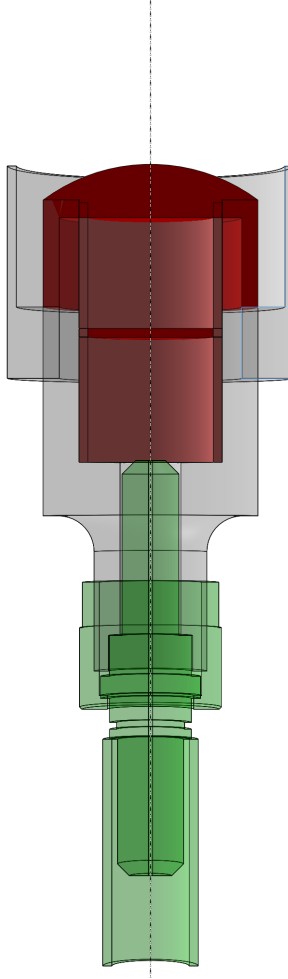

**Figure 7.** Section of our light collectors. The red parts are in teflon, the remaining is in metal. The tip of the fiber (SMA connector) is shown in green.

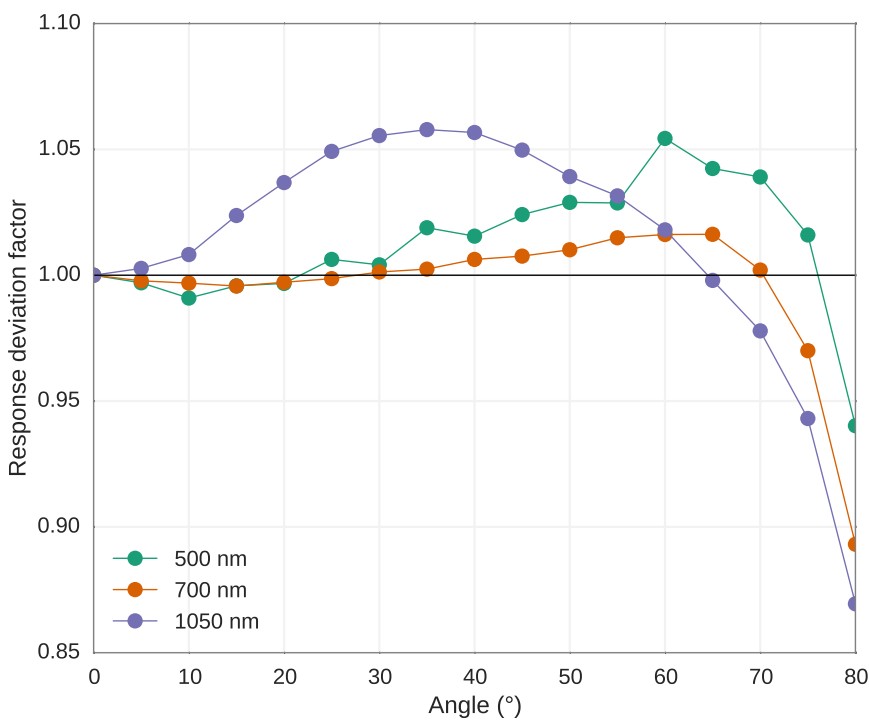

**Figure 8.** Response of the light collector as a function of the angle for different wavelengths. Values are relative to the ideal cosine response and normalized at $0°$.

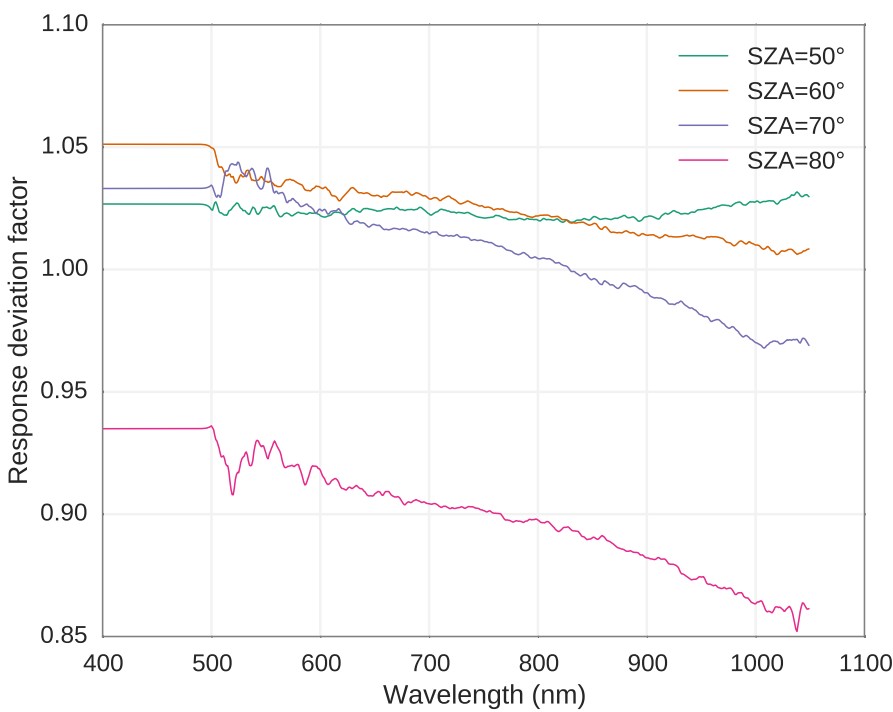

**Figure 9.** Response of the light collector as a function of the wavelength for different incidence angle. Values are relative to the ideal cosine response and normalized at $0°$.

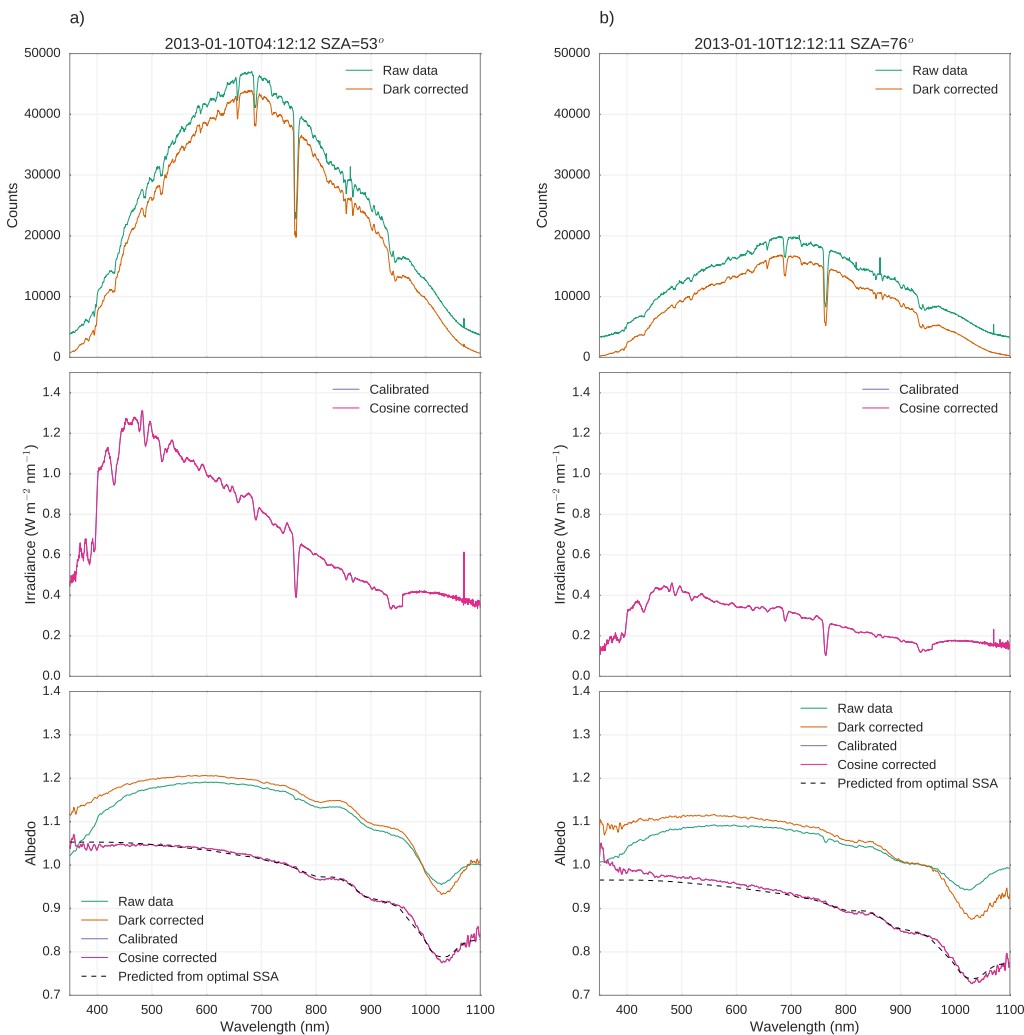

**Figure 10.** Incident spectrum and albedo at different stages of the processing for the 10 January 2013, (left) at 12h12 local time and (right) at 20h12 local time. Raw data (green) are first corrected for the dark current and stray light (orange) then calibrated (violet) and last corrected for imperfect collectors (pink). The theoretical spectrum that fits the fully corrected spectrum (pink) is shown in black.

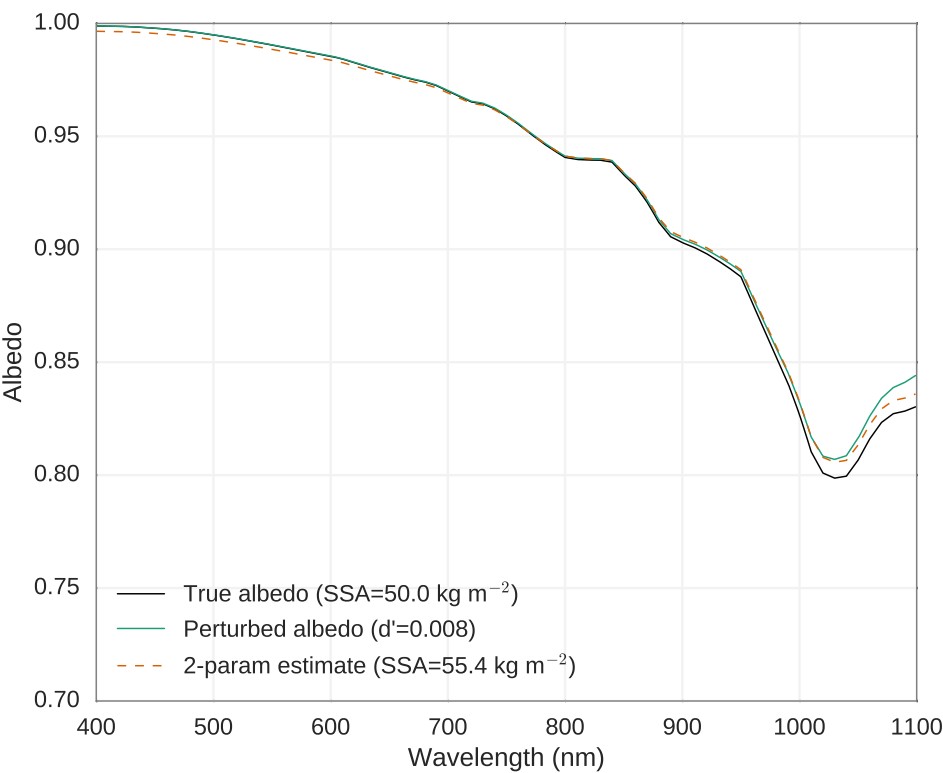

**Figure 11.** Simulations of albedo spectrum for a SZA=60°. A perfect albedo spectrum for an homogeneous snowpack with SSA of 50 $m^2 \, kg^{-1}$ (black) is perturbed (green) by adding an offset of $d' = 0.8\%$ of the maximum incident irradiance to the incident and reflected irradiance. and used as an observation to estimate SSA using the range 700–1050 nm. The estimated spectrum is shown in orange, long dash.

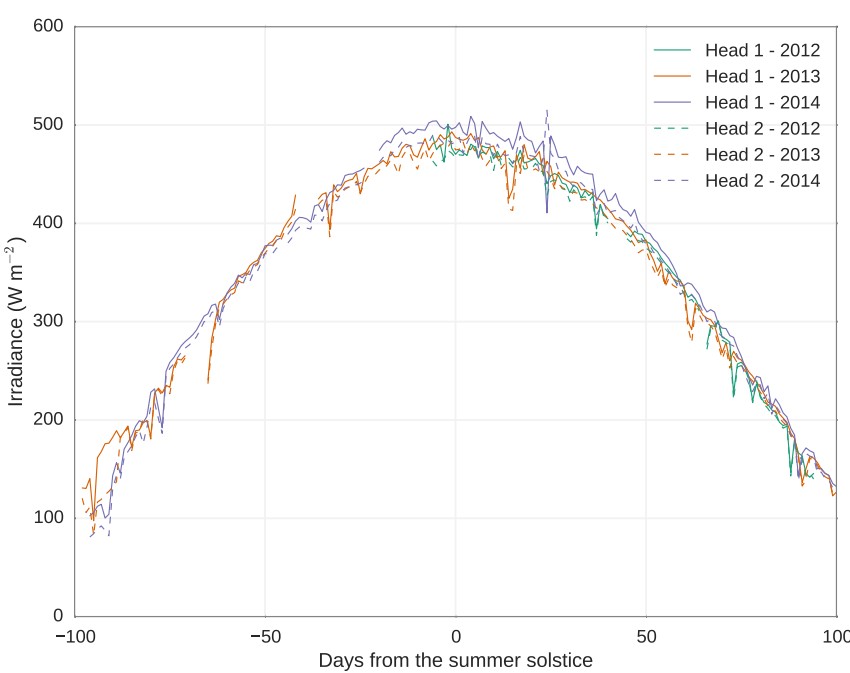

**Figure 12.** Temporal variations of downwelling irradiance measured by the two heads, averaged every day between 12h and 14h local time, and average between 500 and 600 nm for three summer seasons (the year indicated in the legend is that of December)

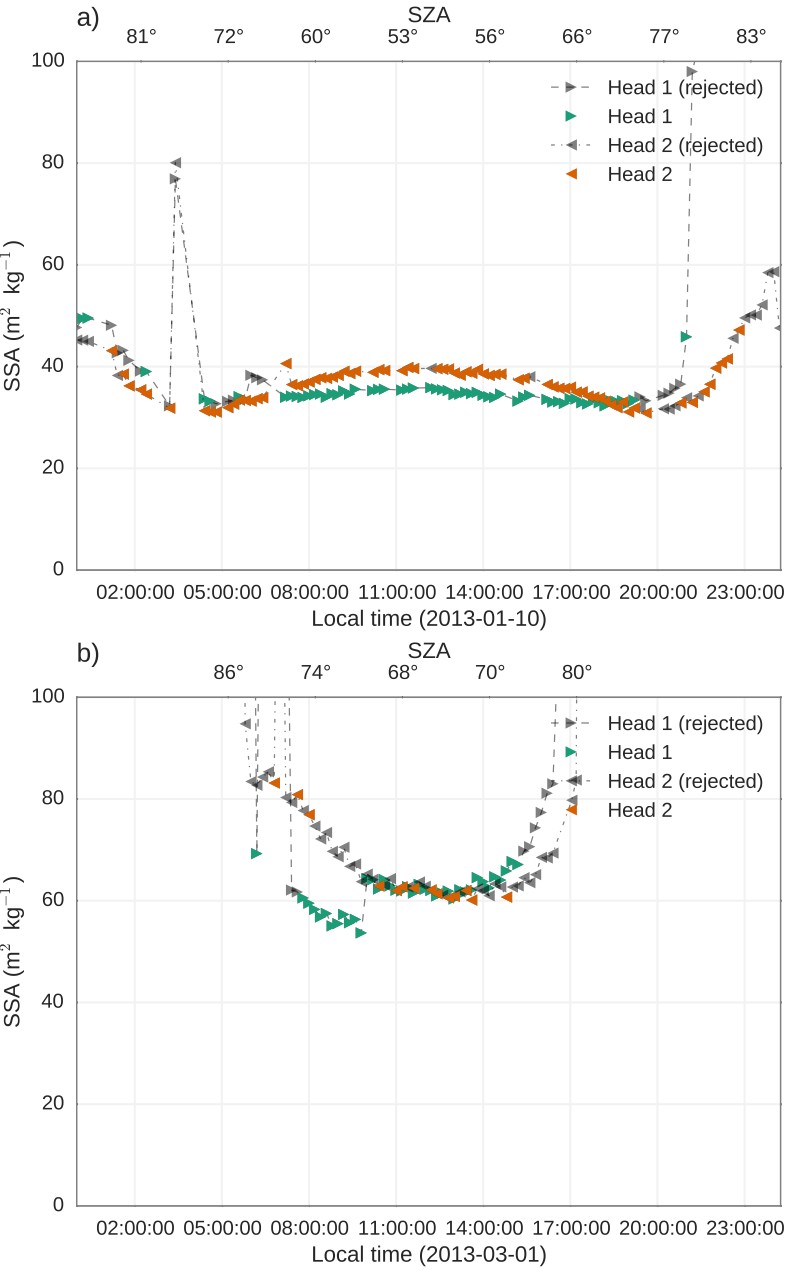

**Figure 13.** Diurnal variations of estimated SSA for 10 January 2013 (top) and 1st March 2013 (bottom) for the two heads. Gray symbols show data rejected by the quality checks.

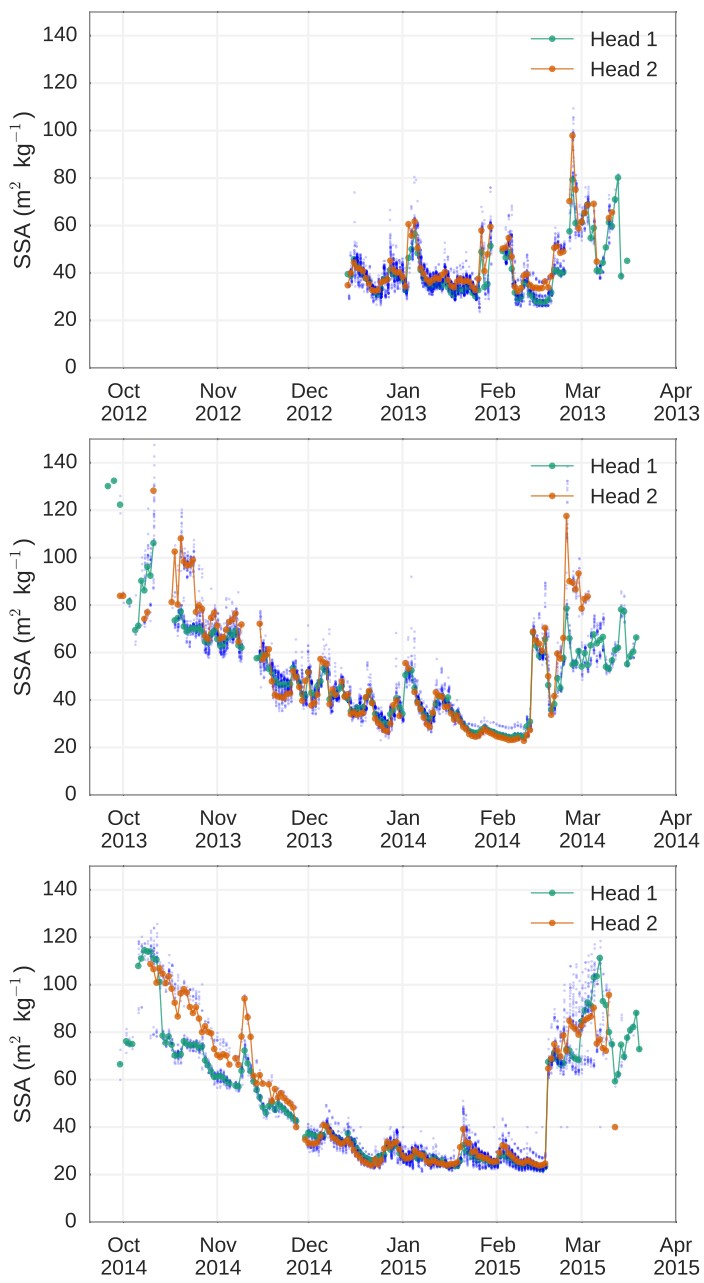

**Figure 14.** Seasonal variations of SSA for three summers since December 2012 for the two heads. The blue dots show individual valid data (both heads) and the symbols show the daily mean of valid data for each head.

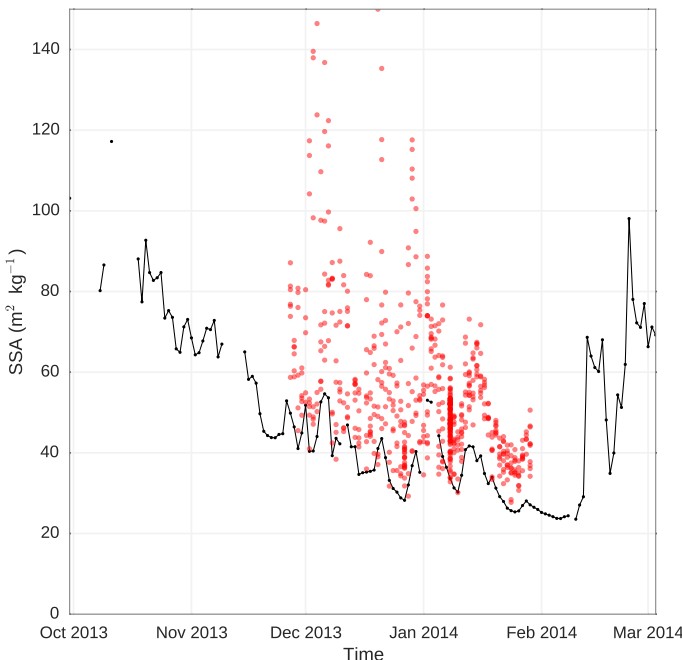

**Figure 15.** Time-series of daily mean SSA averaged for the two heads of Autosolexs (black line) and manual measurements taken using ASSSAP (red dots). Both instruments have different vertical sensitivity as illustrated in Figure 1.