# Peer review of "Development and calibration of an automatic spectral albedometer to estimate near-surface snow SSA time-series"

_The Cryosphere, 2015_

## Referee Comment (RC1) · T. Aoki (Referee) · 17 Feb 2016

This paper presents a method to retrieve the specific surface area (SSA) of snow from spectral albedo measured at Dome-C, Antarctica with an automatic spectrometer "Autosolex". To measure the spectral radiant solar flux with Autosolex many correction procedures are discussed in which error analyses for spectral albedo and SSA were conducted. The diurnal cycle and seasonal variation of SSA are also discussed with possible reasons of these variations.

This manuscript is well-written and addresses the issues on the possible error of SSA in case of SSA retrieval from spectral albedo measurements on the Antarctic Plateau. The descriptions of the instrument are careful and the error analyses are appropriate.

[Figure]

The observed SSA variation is informative and this kind of continuous measurement under such hard circumstance is important to monitor the climate change in Antarctica. I recommend this paper to be published after technical and minor revisions of the following issues.

Specific comments:

p. 3, L19: Typo "cop" -> "cope"

p. 4, L2: "The spectral albedo of the surface is measured" should be preterit.

p. 4, L2-5: The terms of "section" appear here. Do these "sections" differ from sections mentioned at the end of 1. Introduction? The term "subsection" is better.

p.4, L22: Does the sentence "the second one is dedicated to the ultraviolet" explain the right-hand side spectrometer in Fig.2? It is written as "Spectrometer 700-900 nm" in the figure.

p.5, L16-17: "This arises because both the light collector materials and the sky depend on scattering intensity which usually decreases with longer wavelengths." This sentence is a bit difficult to understand particular for "sky". What is it about the sky depend on scattering intensity?

p.7, L10-13: "To estimate the stray light, we assume it affects all the pixels equally ..." Stray light could sometimes cause large error for this kind of instrument. Please show the fraction of contribution from the stray light to "dark and stray light" here or at Fig. 7.

p.8, L17: Equation (6) differs from that of Grenfell et al. (1994) in which the correction for diffuse component is applied. I believe this correction would improve the accuracy of the measured diffuse component.

p.11, L1: "the small peaks due to damaged pixels like at 862 nm and 1069 nm." The small peak can be seen around 862 nm but not at 1069 nm in Fig. 7.

p.11 and Figure 7: The graph names "graph b)" (L11), "graphs (c)" (L17), "graph b"

(L2), and "graph (c)" (L24) are used in the document. However, in Fig. 7 "a)" and "b)" are shown above the top panels, and "(c)" is not indicated. The "graph b)" in L13 seems to discuss on the middle panels.

p.12, L13: The left term of Equation (11) is better "alfa(wavelength)" (as a function of wavelength).

p.14, L19: Equation (16) is not shown.

p.15, L25: "is shown in gray in the background" It is written as "blue dots" in the caption of Fig. 13.

p.15, L27: "the geophysical features to (Libois et al., 2015)." Some terms may disappear before "(Libois et al., 2015)"?
* * *

---

## Referee Comment (RC2) · Anonymous Referee #2 · 29 Mar 2016

General comments: The paper is a breakthrough in the observation of long-term surface properties of snow and spectrally resolved albedo over snow. In my view, the paper was not easy to read, as the technical details specific to albedometers and the snow measurements appear Methods, Results and Discussion - a structure which seems to me not very friendly for the reader. A structure where the instrument details, then the snow measurements, and then a discussion would be easier to follow. I suggest to reorganize the paper into two main sections: - Theory, background, instrumental details - Observations, interpretation and comparison

I also found the title not very relevant, a title as "Development and long-term calibration of a new albedometer measuring vertically integrated snow surface SSA " would be

much more clearly describe what you can expect in this paper.

A second important point concerns the "vertically integrated SSA" (my term). As is clearly shown in Fig. 15, the SSA determined via a spectrally resolved albedometer is in a complex way vertically integrated. As a snowpack is mostly horizontally layered, and often with very strong changes in surface SSA within millimeters (surface hoar, glaze, windcrusts, new snow,...) this point should be considered already in section 2, not in the very end of the paper.

I also found it a bit disturbing that no independent direct measurements of the SSA via stereology or micro-CT were performed. This would make the interpretation significantly more plausible and less speculative, especially concerning the role of the very surface.

As a very significant paper, I would like to suggest to the authors that they reconsider the structure.

As a final point, the peer-review will not be able to validate the method (see p20, L8), I can only check if the methods and procedures are reasonable!

—- Further points p 1 L10 The sentence "The comparison of the retrieved SSA with independent measurements made with an optical device operating at 1310 nm confirms the presence of a sharp and recurrent vertical gradient of SSA in the uppermost centimeter at Dome C, which challenges the assessment of the absolute accuracy from independent measurements." seems to me overstated. If the gradient is "sharp" can not be determined by the methods used: either near-infrared photography in a profile would be necessary, or micro-tomography. What is obvious and correct that the upper snow layers are at times of higher SSA than the averaged SSA observed with Autosolex. This is not surprising concerning the calculated penetration depth! The same is valid for p17 L34ff.

p 3 L 27 ff references for the "manual devices"?

p18 L11 ff: Which spectral irradiance was used (W m-2 nm-1) for the calculation of the averaged penetration depth?

p22 L19 reference to Libous 2013 seems incomplete

p38 Fig. 14 The two datasets are from different depths (as shown in Fig. 15), so they are not really comparable. These data only show that the surface has almost always a higher SSA than the deeper "sampling" Autosolex.

---

## Author Response (AR1)

**Review 1**

This paper presents a method to retrieve the specific surface area (SSA) of snow from spectral albedo measured at Dome-C, Antarctica with an automatic spectrometer "Autosolex". To measure the spectral radiant solar flux with Autosolex many correction procedures are discussed in which error analyses for spectral albedo and SSA were conducted. The diurnal cycle and seasonal variation of SSA are also discussed with possible reasons of these variations.

This manuscript is well-written and addresses the issues on the possible error of SSA in case of SSA retrieval from spectral albedo measurements on the Antarctic Plateau. The descriptions of the instrument are careful and the error analyses are appropriate. The observed SSA variation is informative and this kind of continuous measurement under such hard circumstance is important to monitor the climate change in Antarctica. I recommend this paper to be published after technical and minor revisions of the following issues.

We are grateful to the reviewer for his positive remarks and have made modifications in the manuscript according to his recommendations.

Specific comments:
p. 3, L19: Typo "cop" -> "cope"

done

p. 4, L2: "The spectral albedo of the surface is measured" should be preterit.

done

p. 4, L2-5: The terms of "section" appear here. Do these "sections" differ from sections mentioned at the end of 1. Introduction? The term "subsection" is better.

done

p.4, L22: Does the sentence "the second one is dedicated to the ultraviolet" explain the right-hand side spectrometer in Fig.2? It is written as "Spectrometer 700-900 nm" in the figure.

Both were erroneous, ultraviolet is infrared and the range in Fig.2 is 700-1100nm. This is corrected.

p.5, L16-17: "This arises because both the light collector materials and the sky depend on scattering intensity which usually decreases with longer wavelengths." This sentence is a bit difficult to understand particular for "sky". What is it about the sky depend on scattering intensity?

We change to "This arises because scattering by the light collector materials and the atmosphere strongly decreases with longer wavelengths."

p.7, L10-13: "To estimate the stray light, we assume it affects all the pixels equally ..." Stray light could sometimes cause large error for this kind of instrument. Please show

the fraction of contribution from the stray light to "dark and stray light" here or at Fig. 7.

We have separated the two corrections in the plot below (similar to Fig.7) and found the curve after the stray light correction (violet) can not be distinguished from the curve (orange) after the dark correction which means the stray light is very small. Instead of adding a figure with overlapping curves, we first show and address the first step alone (dark correction) and then add a short paragraph on the stray light correction "The next step (stray light correction) is not shown in Figure 7 because it has a negligible effect."

[Figure]

p.8, L17: Equation (6) differs from that of Grenfell et al. (1994) in which the correction for diffuse component is applied. I believe this correction would improve the accuracy of the measured diffuse component.

The equation (6) is indeed formally different from that given in Grenfell et al. (1994) because our absolute calibration is done with diffuse radiation and the imperfect cosine. It means the diffuse component is already calibrated after the "absolute calibration step". Despite this formal difference, considering two steps "absolute calibration"+"cosine correction" together is equivalent to Grenfell et al. We have added an explanation: "The (cosine) correction is applied only to the direct component as the diffuse component has been already calibrated at the absolute calibration step"

p.11, L1: "the small peaks due to damaged pixels like at 862 nm and 1069 nm." The small peak can be seen around 862 nm but not at 1069 nm in Fig. 7.

The x-axis scale of Figure 7 has been extended to show the peak around 1069 nm.

p.11 and Figure 7: The graph names "graph b)" (L11), "graphs (c)" (L17), "graph b" (L2), and "graph (c)" (L24) are used in the document. However, in Fig. 7 "a)" and "b)" are shown above the top panels, and "(c)" is not indicated. The "graph b)" in L13 seems to discuss on the middle panels.

We have corrected this: letters are used to designate the columns and "top, middle and bottom" are now used for the graphs within a column.

p.12, L13: The left term of Equation (11) is better "alfa(wavelength)" (as a function of wavelength).

Yes, we have also added the angle dependence.

p.14, L19: Equation (16) is not shown.

This blank equation was due to a latex typo,

p.15, L25: "is shown in gray in the background" It is written as "blue dots" in the caption of Fig. 13.

"gray" changed to "blue".

p.15, L27: "the geophysical features to (Libois et al., 2015)." Some terms may disappear before "(Libois et al., 2015)"?

The parenthesis have been removed and the sentence changed to :
" The interpretation of the geophysical features is addressed in Libois et al. 2015."

**Review 2**

General comments: The paper is a breakthrough in the observation of long-term surface properties of snow and spectrally resolved albedo over snow. In my view, the paper was not easy to read, as the technical details specific to albedometers and the snow measurements appear Methods, Results and Discussion - a structure which seems to me not very friendly for the reader. A structure where the instrument details, then the snow measurements, and then a discussion would be easier to follow. I suggest to re-organize the paper into two main sections:

- Theory, background, instrumental details

- Observations, interpretation and comparison.

We have reorganized the paper to address first the most generic and theoretical points and second the instrument details. After the introduction, a section addresses the theory  with 1) a description of the SSA retrieval algorithm, 2) the vertical representativeness of the retrieved SSA and 3) the formulation to investigate the impact of instrument artifacts on SSA uncertainty. The next section is focused on the instrument description, processing description and illustration/uncertainties, and stability. The next section presents the time-series of SSA and the last one provides conclusions and future works. This follows the two points proposed by the reviewer except that the "Theory" and "instrumental details" are split in two sections because they are indeed different and represent 90% of the article.

I also found the title not very relevant, a title as "Development and long-term calibration of a new albedometer measuring vertically integrated snow surface SSA " would be much more clearly describe what you can expect in this paper.

We propose the following title which include the term "time-series" which is the main novelty of the instrument.

"Development and calibration of an automatic spectral albedometer to estimate near-surface snow SSA time-series."

A second important point concerns the "vertically integrated SSA" (my term). As is clearly shown in Fig. 15, the SSA determined via a spectrally resolved albedometer is in a complex way vertically integrated. As a snowpack is mostly horizontally layered, and often with very strong changes in surface SSA within millimeters (surface hoar, glaze, windcrusts, new snow,...) this point should be

considered already in section 2, not in the very end of the paper.

This has been moved in Section 2 as suggested.

I also found it a bit disturbing that no independent direct measurements of the SSA via stereology or micro-CT were performed. This would make the interpretation significantly more plausible and less speculative, especially concerning the role of the very surface.

We share this feeling but we do not have such independent measurements at Dome C and we believe it is very difficult to collect near surface snow and apply X-ray tomography (or stereology) techniques to obtain results with a sufficient accuracy in the typical Dome C conditions of high SSA and high vertical gradient in the topmost layer.

We have participated in several intercomparison campaigns over the past years where many measurements techniques have been combined. These intercomparisons took place in Alpine conditions where SSA is lower, snow is more homogeneous and the vertical gradient is probably much smaller than at Dome C. Only in such ideal conditions and after a few attempts, we were able to claim that a reasonable check (or "cross-validation") of the optically-based SSA measurements has been achieved (not published yet). We hope such attempts could be done at Dome C in the future.

As a very significant paper, I would like to suggest to the authors that they reconsider the structure.

As a final point, the peer-review will not be able to validate the method (see p20, L8), I can only check if the methods and procedures are reasonable!

Yes. The sentence is removed.

- Further points p 1 L10 The sentence "The comparison of the retrieved SSA with independent measurements made with an optical device operating at 1310 nm confirms the presence of a sharp and recurrent vertical gradient of SSA in the uppermost centimeter at Dome C, which challenges the assessment of the absolute accuracy from independent measurements." seems to me overstated. If the gradient is "sharp" can not be determined by the methods used: either near-infrared photography in a profile would be necessary, or micro-tomography. What is obvious and correct that the upper

snow layers are at times of higher SSA than the averaged SSA observed with Autosolex. This is not surprising concerning the calculated penetration depth! The same is valid for p17 L34ff.

We remove this sentence from the abstract because the paper does not address the issue of the gradient with sufficient details.

p 3 L 27 ff references for the "manual devices"?

done

p18 L11 ff: Which spectral irradiance was used (W m-2 nm-1) for the calculation of the averaged penetration depth?

Albedo calculations from which we deduce the penetration depth are independent of the irradiance.

p22 L19 reference to Libous 2013 seems incomplete

corrected

p38 Fig. 14 The two datasets are from different depths (as shown in Fig. 15), so they are not really comparable. These data only show that the surface has almost always a higher SSA than the deeper "sampling" Autosolex.

This has been addressed with the reogranisation of the article. The vertical representativeness is first described in Section 2. Then the comparison with Asssap in Section 4 only concludes that the SSA is always higher at the surface than slightly deeper.

[revised manuscript text omitted]

**3.4**

20 ~~the number of observables per spectrum is very large (up to 2000 different wavelengths), the errors and the auto-correlation between these observations require to keep the number of unknown low to obtain a stable fit. The following assumptions are made: 1) The snowpack is horizontally and vertically homogeneous which means only one SSA value is retrieved. 2) The surface is flat. Roughness which tends to smooth the solar zenith angular response of the snow Warren et al. (1998) is neglected. 3) The surface and the sensor are horizontal. 4) Snow phase function and single scattering albedo are implicitly described by~~
25

30 $$\alpha^{\mathrm{diff}}(\lambda) = \exp\left(-4\sqrt{\tfrac{2B\gamma_\lambda}{3\rho_{\mathrm{ice}}\mathrm{SSA}(1-g)}}\right)$$

$$\alpha^{\mathrm{dir}}(\lambda,\theta) = \exp\left(-\tfrac{12}{7}(1+2\cos\theta)\sqrt{\tfrac{2B\gamma(\lambda)}{3\rho_{\mathrm{ice}}\mathrm{SSA}(1-g)}}\right),$$

where $\theta$ is the solar zenith angle, $\rho_{\mathrm{ice}} = 917$ is the ice density at $0$, and $\gamma(\lambda)$ is the absorption coefficient of ice, taken from Warren and Brandt (2008). $B = 1.6$ and $g = 0.85$ are respectively the absorption enhancement and the asymmetry factor values suggested by Libois et al. (2014b).

Considering both direct and diffuse radiations as in Equation (17), the model reads:

$$\alpha^{\text{1-param}}(\lambda, \theta) = \left[ r^{\text{diff}}(\lambda, \theta) \alpha^{\text{diff}}(\lambda) + \left( 1 - r^{\text{diff}}(\lambda, \theta) \right) \alpha^{\text{dir}}(\lambda) \right],$$

where $r^{\text{diff}}(\lambda, \theta)$ is calculated as for the cosine correction. This model is called hereinafter 1-parameter model because only the SSA is unknown. However, it is not the most suitable because the measurements are often affected by errors which result in a slight wavelength-independent scaling of the albedo. Typical examples include the variations of the illumination between the two successive acquisitions of downwelling and upwelling irradiance, the effect of sloping surface and the imperfection of the cross-calibration. To cope with these errors without negatively impacting the SSA estimation, another model including a free scaling parameter $A$ is also used:

$$\alpha^{\text{2-param}}(\lambda, \theta) = A \left[ r^{\text{diff}}(\lambda, \theta) \alpha^{\text{diff}}(\lambda) + \left( 1 - r^{\text{diff}}(\lambda, \theta) \right) \alpha^{\text{dir}}(\lambda) \right],$$

Nevertheless, considering that a value of $A$ significantly different from 1 reveals a serious issue with the observations, we reject any fit that gives $A$ outside the range $0.9 - 1.1$.

The fit is performed with a non-linear least square method (provided by the Python scipy.optimize.leastsq function) to minimize the squared difference between the model and observed spectrum. Smoothing is done beforehand to Before applying the SSA retrieval algorithm, albedo spectra are smoothed to reduce the noise using a first order Butterworth low-pass provided by the Python scipy.signal.butter function with cut-off of 0.1.

Given the stronger sensitivity of the albedo to SSA in the near-infrared with respect to the visible, only the data for wavelengths in the range $700 - 1050$ are used for the fit. This choice is a compromise. Extending the range towards the shorter wavelengths does not bring useful information for the SSA retrieval and may add artifacts, while narrowing it would reduce the number of observations and result in a greater sensitivity to noise. Extending the range in the longer wavelengths is not possible due to the weak sensitivity of the spectrometer.

**4 Results**

**3.1 StabilityIllustration of the processing steps**

Before deriving albedo and SSA over long periods of time, it is important to check the overall stability of the instrument. The seasonal variations of downwelling irradiance are plotted in Figure **??** for the two heads and for the three summer seasons. The irradiance is integrated between 400 and 1000 and averaged between $10\mathrm{h} - 14\mathrm{h}$ (local time). These data include all weather conditions. Overall, there is a good year to year agreement. For instance, the irradiance observed by the head 1 during the 30 days after the solstice (period in common for the three seasons) has increased by +0.5 % and +4.2% the second and third years with respect to the first one. These values become -0.5%, and +3.0% respectively for the head 2, which is relatively

small for an unattended instrument and considering that these values include the inter-annual variability of cloudiness. Only the third season shows a significant increase (around 4-5 %) which is likely due to a degradation of the leveling. The prevailing southerlies winds may have tilted the mast northwards, resulting in the observed increase of the noon irradiances. The influence of tilt angle on the estimation of the SSA is not critical because small geometrical imperfections are partially compensated when using the 2-parameter model that includes the scaling factor $A$. This influence is evaluated in Section 2.3.2 but we do not know the actual tilt of the heads because the inclinometers mounted on the system appeared to be deficient at low temperature.

**3.2 Measurement processing**

[revised manuscript text omitted]

 The collector angular response correction (from violet to pink curves) has very small impact on the radiance (middle graph) whatever the acquisition hour. The impact is also apparently moderate on the albedo (bottom graph). The correction factor ranges from 0.98 in the blue to 0.99 in the near infra-red at noon and 1.00 to 1.07 at 8 pm. Even if these values are weak (maximum of 7%), the correction succesfully removes the decreasing trend in the shorter wavelengths of the visible (400 to 600 nm) that affects the evening acquisition. Albedo measurements by Nicolaus et al. (2010) show a similar artifact. Such a trend is visible throughout the timeseries at large SZA and the collector response correction usually performs well by recovering a nearly constant value as expected.

Regarding SSA estimation, the correction has a little impact for the noon acquisition (36.7 and 36.2 $\mathrm{m^2\,kg^{-1}}$ respectively before and after the correction) as expected. In contrast, for the evening acquisition the SSA estimate increases from 20 $\mathrm{m^2\,kg^{-1}}$ to 35 $\mathrm{m^2\,kg^{-1}}$, the latter being close to the value at noon. This clearly shows that the correction of the collector response is crucial. The theoretical spectrum that fits the fully-corrected spectrum is shown in black in Figure **??**. The differences between the fit and the observation are small, we can note a slight over-estimation at 800 nm and under-estimation at 970 nm which are very likely due to a small error in the refractive index of the ice already pointed out in other studies (e.g. Carmagnola et al., 2013). In contrast, the over-estimation at 1030 nm is more likely an error in the observations resulting from the low sensitivity of the spectrometer in this wavelength range.

To further investigate the impact of the albedo uncertainties due to the instrument artifacts on the retrieved SSA , we use numerical experiments.

**3.2 Stability**

**3.3 Numerical evaluation of the uncertainty**

5 Most artifacts affecting the spectral radiometer results in a first approximation in either an offset in the radiances or a chromatic aberration. To quantify the respective effects of these artifacts on the estimation of the SSA, we perform simple and idealized modeling experiments as follows: we first compute the perfect albedo spectrum with the model described in Equation 18 for a given SSA (called true SSA) and perturb it to mimic the imperfect response Before deriving albedo and SSA over long periods of time, it is important to assess the overall stability of the instrument. Then, we estimate the SSA using the same procedure as
10 used with real data and deduce the error as the difference between the estimated and true SSA.

**3.2.1 Offset**

To study the effect of the dark current and stray light correction, we model the measured albedo as:

$$\alpha = \frac{S^{\text{ref}}(\lambda) + d}{S^{\text{inc}}(\lambda) + d},$$

where the true reflected and incident radiance spectra $S^{\text{ref}}(\lambda)$ and $S^{\text{inc}}(\lambda)$ are affected by an offset $d$. Dividing by the incident
15 radiance to make explicit the true albedo $\alpha^{\text{true}}$ yields:

$$\alpha(\lambda) = \frac{\alpha^{\text{true}} + \frac{d}{S^{\text{inc}}(\lambda)}}{1 + \frac{d}{S^{\text{inc}}(\lambda)}}.$$

For $S^{\text{inc}}(\lambda)$ we chose a Gaussian shape looking like the raw observations in Figure ??:

$$S^{\text{inc}}(\lambda) = S_{\text{mode}} \exp\left[-\left(\frac{\lambda - 680\text{nm}}{270\text{nm}}\right)^2\right].$$

The amplitude of the spectrum $S_{\text{mode}}$ is usually of the order of the resolution of the Digital Analog Converter of the spectrometer
20 ($2^{16}$ in our case) if the integration time is optimal. We define $d' = d/S_{\text{mode}}$, the relative contribution of the offset with respect to the spectrum amplitude. We then estimate the SSA from the perturbed $\alpha(\lambda)$ spectrum and calculate the difference with respect to the true SSA.

To apply this theory to assess the impact of an imprecise dark and stray light correction, we take typical values from the noon acquisition studied in Section 3.2 : the spectrum has an amplitude of nearly 50000 and the offset is around 3800. We further
25 assume that this offset can estimated with an accuracy of 10%, which results in a residual bias of 380 and thereby $d' = 0.8\%$. Figure ?? shows the perfect albedo (black curve) calculated for a true SSA of 50 and the perturbed albedo (green curve) with

5 ~~independent of the true SSA. To meet the 15% criteria, the offset due to dark current and stray light should be corrected with an accuracy better than $d'$=1%. In theory, such an accuracy can be achieved using a single measurement of one dark pixel as our spectrometer feature a signal-to-noise ratio of 1:500. By averaging many dark pixels, as we do, we ensure an even better accuracy meaning that the offset correction is not an issue in our case. Nevertheless, it is worth noting that we assume the stray light effect to be uniform on all the pixels of the CMOS sensor. If this is not the case, the correction of~~ The leveling is a first and
10 important parameter. It was carefully done at the installation of Autosolexs in December 2012 within the uncertainty allowed by spirit level (about 0.2°) but could have degraded. Nevertheless, in December 2015, new measurements with an electronic devices yielded insignificant movement of  structure.

**3.2.1**

15

$$\alpha = \left(1 - b\frac{\lambda - \lambda_0}{\lambda_1 - \lambda_0}\right)\alpha^{\mathrm{dir}}(\lambda, \theta)$$

The radiometric stability is another parameter. The seasonal variations of downwelling irradiance are plotted in Figure **??** for the two heads and for the three summer seasons. The
20 irradiance is integrated between 400 and  1000 nm  and averaged between 10h –
25

14h (local time). These data include all weather conditions. Overall, there is a good year to year agreement. For in-
30 stance, ~~in Figure **??**, the difference is 0.024 on average over the range $400 - 550$ . This difference (residuals hereinafter) is a consequence of the wavelength-dependent perturbation and can be exploited to assess the amplitude of this perturbation in real data because it does not require knowledge of the true SSA. It means that the acquisitions subject to residual chromatic aberration should feature a large difference in the visible range with respect to the spectrum predicted using the 2-parameter~~

model. The previous calculation with $b = 0.05$ and true SSA=50 gives a difference of 0.024 and was found to corresponds to relative error of 24% of SSA. To explore how these values vary in other conditions, we perform several simulations. Figure ??a shows the relative error and the mean difference over the range $400 - 550$ as a function of the chromaticity parameter $b$ for several SSA. The error varies almost linearly with $b$ and the slope increases with SSA. Hence, to attain the target accuracy of

5    15%, the chromatic parameter should not exceed about 0.04 for SSA=20 and even less, 0.02, for SSA=100. Figure ??b shows the univocal relationship between $b$ and the mean residual. This result suggests that mean residual values under 0.01 indicate a chromaticity better than 0.02 and thus an SSA accuracy better than 15%.

We apply this criteria on the real data by calculating the albedo spectrum predicted by 2-parameter model using the optimal SSA and $A$ obtained by fitting the model to the observations in the range $700 - 1050$. We then calculate the mean difference

10    in the range $400 - 550$ and reject the acquisition if this difference is larger than 0.01. The skills of this filter are illustrated in the next section. It is worth noting that it can only work if the light-absorbing impurities are negligible because dust or black carbon usually lead to a decrease of the albedo in the blue-green (Warren, 1982) which would be interpreted as an artifact by our filter that is based on pure snow albedo calculation.

It is interesting to estimate the chromatic aberration associated with the different steps of the processing or other external

15    sources of error. For instance, the cross-calibration factor ranges from 0.95 in the blue to 0.85 in the infrared (Section 3.2) which corresponds to approximately $b = 0.10$. It means that the cross-calibration is definitely required to reach the 15%accuracy. If we assume that the correction factor is accurate within 10%, for instance because of the limited reproducibility of the experiment to determine the reference spectra, it leads to a residual trend of $b \approx 0.01$ which is weak enough to reach the 15% accuracy. A similar conclusion can be drawn for the collector angular response correction that resulted in similar scaling factors, between

20    1.00 to 1.07 for the blue and near infra-red wavelengths respectively for the evening acquisition studied in Section 3.2 and much lower factors for the noon acquisition.

Errors of the leveling of the measurement heads also indirectly results in chromatic aberration because it affects the direct/diffuse balance. Following ? and Grenfell et al. (1994), the albedo error $\eta(\lambda, \theta)$ due to a tilt angle $\delta$ in the direction of the sun (worse case) can be written:

25    $$\eta(\lambda, \theta) \approx \left(1 - r^{\mathrm{diff}}(\lambda, \theta)\right)\left(\frac{\cos(\theta - \delta)}{\cos(\theta)} - 1\right)$$

where the diffuse term error has been neglected as suggested by ? results. Considering small errors, $b$ can be estimated as follows:

$$b \qquad\qquad \approx \eta(\lambda_1, \theta) - \eta(\lambda_0, \theta)$$

30    $$\approx \left(r^{\mathrm{diff}}(\lambda_1, \theta) - r^{\mathrm{diff}}(\lambda_0, \theta)\right)\left(\frac{\cos(\theta - \delta)}{\cos(\theta)} - 1\right)$$

For a tilt angle of the irradiance observed by the head 1 and under Dome C typical conditions, we obtain $b = 0.01$ for SZA=53 (as the noon acquisition studied in Section 3.2) and $b = 0.05$ for SZA=77 (as the evening acquisition) which is weak in the former case but is too high in the latter one to reach the 15% target accuracy. These results highlight the crucial role of

the leveling of the instrument. When, Autosolexs was installed, the leveling was carefully done within the uncertainty allowed by spirit level (about 0.2) but it is likely that it has degraded over time. Given that measuring tilt angle with electronic devices in Dome C winter conditions is challenging and that the instrument was not serviced to preserve the surface, the confidence in our data is based on the *ad hoc* filtering based on the 2-parameter model residual in the visible. during the 30 days after the solstice (period in common for the three seasons) has increased by +0.5 % and +4.2% the second and third years with respect to the first one. These values become -0.5%, and +3.0% respectively for the head 2, which is relatively small for an unattended instrument and considering that these values include the inter-annual variability of cloudiness. Only the third season shows a significant increase (around 4-5 %) which is unexplained.

The effect of the different artifacts has been evaluated independently of each other, but in real data, they combine, sometimes compensating, sometimes not. The assessment of the interactions between the artifacts is difficult and is not addressed here, but reaching 15% accuracy in real data may require stricter tuning of the filters

[revised manuscript text omitted]
. Carmagnola et al. (2013) at Summit Greenland uses spectral data in the wavelength range encompassing Autosolexs and ASSSAP operating ranges and found that tuning the surface layer SSA helps to improve the agreement with the observations but did not explore values as high as Grenfell et al. (1994) did. In contrast, Gallet et al. (2011) measured SSA vertical profiles at Dome C and obtained radiative transfer calculations in agreement with Hudson et al. (2006) observations without adding such a layer. Using snowpack numerical modeling, Libois et al. (2015) obtained SSA profiles at Dome C with a sharp gradient near the surface. It is worth noting that their simulation~~ large difference in the first part of the season can be attributed to the frequent blowing snow events occurring during this period but the difference persists over 8 January despite calmer conditions. This persistence of large difference of SSA over a small vertical distance is not well understood (Gallet et al., 2011). Clear-sky precipitation (a.k.a diamond dust) is a possible candidate but snow evolution simulation performed by Libois et al. (2015) uses ERA-Interim as input. The reanalysis successfully forecasts the precipitation events which leads to marked rises of SSA (Figure ??) but does not  include diamond dust. Since the simulation agrees with the SSA observations, this indicates that diamond dust is not required in the model to get a sustained high surface SSA  on the surface. We can conclude that a  significant SSA gradient in the first uppermost centimeters exists with SSA possibly exceeding 100

$\mathrm{m^2\,kg^{-1}}$ on the very surface and that this gradient persists over long periods, even during periods without precipitation and blowing snow.

As a consequence, the vertical representativeness of the SSA estimated in this study is an important information in order to analyze the data and compare them to models or other observations. It is therefore important to determine the thickness of the

5   layer sampled by Autosolexs. The answer is not straightforward nor univocal because the penetration depth of radiation depends on the wavelength and the vertical profile of snowproperties, including SSA itself and density. Furthermore, the estimated SSA value is not a simple average over a given thickness but is weighted by a decreasing function of the depth. This function is exponential if the snow is homogeneous, but more complex otherwise.

Only radiative transfer calculations can give a precise estimation of the weight function. To evaluate it for an homogeneous

10   snowpack, we run the two-stream radiative transfer model TARTES (Libois et al., 2013) considering a semi-infinite medium with a density of 270 corresponding to mean surface conditions (Libois et al., 2014a) and a fixed SSA. A layer with variable thickness $h$ is added on top of it, with the same properties, excepted that a small perturbation is added to the SSA (e.g. +1 , but the exact value does not alter the final result as long as it is small compared to the actual value of SSA). The relative contribution of the uppermost layer to the albedo is then defined as the quantity $(\alpha(h) - \alpha(0))/(\alpha(\infty) - \alpha(0))$ where $\alpha(h)$ is

15   the albedo calculated by TARTES for the thickness $h$ and averaged over the wavelength range used by Autosolexs (700-1050 ). The contribution is shown in Figure **??** (plain line) as a function of $h$ for different SSA values. Results show that the uppermost 10 mm snow layer contributes to nearly 60% of the albedo for SSA of 20 and 85% for higher SSA of 100 . Conversely, the layer contributing to 80% of the signal is 18, 12 and 8thick for SSA of 20, 50 and 100 respectively. As a conclusion, 1 is a good approximation for typical Dome C conditions. Similar simulations have been run at 1310 to illustrate the difference with

20   ASSSAP (dash line in Figure **??**). In this case most of the signal comes from uppermost 5 , at most, in any conditions. Note that this depth is inversely proportional to the density so that the thickness values presented here should be multiplied by about two in case of fresh snow or surface hoar.

The question of the representativeness of the SSA estimated with the instruments is relevant for comparing the observations with snow models like Crocus or to study the snow processes. As far as the broadband albedo and the surface radiative budget

25   are concerned, SSA estimated in the range 700 – 1050 is the most relevant as most of the solar energy is absorbed in the near-infrared range (Gardner and Sharp, 2010).

**5   Discussions and Conclusion**

[revised manuscript text omitted]